# Can Less be More? When Increasing-to-Balancing Label Noise Rates Considered Beneficial

**Yang Liu**
Computer Science and Engineering
University of California, Santa Cruz
Santa Cruz, CA 95064
yangliu@ucsc.edu

**Jialu Wang**
Computer Science and Engineering
University of California, Santa Cruz
Santa Cruz, CA 95064
faldict@ucsc.edu

## Abstract

In this paper, we answer the question of when inserting label noise (*less informative labels*) can instead return us *more* accurate and fair models. We are primarily inspired by three observations: 1) In contrast to reducing label noise rates, increasing the noise rates is easy to implement; 2) Increasing a certain class of instances' label noise to balance the noise rates (increasing-to-balancing) results in an easier learning problem; 3) Increasing-to-balancing improves fairness guarantees against label bias. In this paper, we first quantify the trade-offs introduced by increasing a certain group of instances' label noise rate w.r.t. the loss of label informativeness and the lowered learning difficulties. We analytically demonstrate when such an increase is beneficial, in terms of either improved generalization power or the fairness guarantees. Then we present a method to insert label noise properly for the task of learning with noisy labels, either without or with a fairness constraint. The primary technical challenge we face is due to the fact that we would not know which data instances are suffering from higher noise, and we would not have the ground truth labels to verify any possible hypothesis. We propose a detection method that informs us which group of labels might suffer from higher noise without using ground truth labels. We formally establish the effectiveness of the proposed solution and demonstrate it with extensive experiments.

## 1 Introduction

The presence of training label noise is generally considered harmful. Typically the goal of learning with noisy labels is to improve the training by reducing the amount of noise in the training data [4, 11]. This paper discusses the feasibility of the opposite strategy and shows cases when adding label noise, leading to a scenario with *less* informative labels, will result in *more* accurate and fair models. We are primarily motivated by three observations:

**Observation I: Reducing label noise is hard, but increasing it is easy.** While the literature has provided us with solutions to perform data cleaning, they often require a highly customized training process. Their lack of theoretical rigor has also posed challenges when performing evaluations. On the other hand, as we will see later, increasing label noise is easy — one can always do so by randomly flipping the current noisy labels to increase it further.

**Observation II: Increasing a certain class of instances' noise rate to balance the noise rates results in an easier learning problem.** When label noise is class-dependent (label class $Y = +1$ v.s. $Y = -1$), popular noise-tolerant learning algorithms typically require the knowledge of noise rates [25]. However, the learner often needs to identify the unknown noise rates by some estimation procedure [26, 27, 36]. We articulate that the mis-specification of noise rates will introduce additional learning errors, especially when the label noise is asymmetric (see Theorem 1). On the other

hand, although increasing the noise rate of the lower class one to balanced error rates reduces the informativeness of the training labels, it is regarded as an easier case to handle — when error rates are balanced, invoking loss correction procedures becomes unnecessary (see Lemma 2 & Theorem 3).

**Observation III: Increasing-to-balancing improves fairness guarantees against label bias.** When the label noise rates are group-dependent (data from young group v.s. senior group), it has been reported in the literature that imposing fairness constraints directly on the noisy labels would instead reinforce the unfairness [30]. Fixing the fairness constraints again requires the knowledge of the label noise rates. We will show that increasing and balancing noise rates allows us to directly impose fairness guarantees on the noisy training data without knowing the noise rates (Theorem 6).

This paper first quantifies the trade-offs introduced by increasing a certain group of instances' label noise rate with respect to the loss of informative labels and the gained benefits for doing so. We analytically demonstrate when such an increase proves beneficial in terms of either improved generalization error or fairness guarantees. Then we present a method to leverage our idea of inserting label noise for the task of learning with noisy labels, either without or with a fairness constraint. The primary technical challenge we face is that we would not know which data instances are suffering from higher noise, and again we would not have the ground truth labels to verify any possible hypothesis. In response, we propose a detection method that informs us which group of labels might suffer from higher noise without using ground truth information. The core discovery is a couple of metrics that check the agreements of noisy labels among local neighbors. These two metrics can be easily estimated using noisy labels only and are shown to be sufficient to inform us of the class of labels with a higher noise rate. With this knowledge, we propose an algorithm (NOISE+) to gradually insert noise into the lower noise class of labels. Our contributions summarize as follows:

- Our paper prototypes the idea of pre-processing noisy training labels for both the constrained and unconstrained learning problems. We show the possibility of improving the training accuracy and fairness guarantees by increasing a certain class of instances' noise rates (increasing-to-balancing).

- To enable the deployment of our idea, we propose a detection algorithm to identify the class of instances with a higher label noise rate without using any ground truth label information.

- Our solution contributes to the learning with noisy labels literature by adding another tool that is robust to noise rate (noise transition matrix) estimation errors.

All omitted proofs can be found in the Appendix. The code for reproducing the experimental results is available at https://github.com/UCSC-REAL/CanLessBeMore.

## 1.1 Related work

Our work is a contribution to the well-established literature of learning with noisy labels [11, 22, 25, 27–29, 33]. The classical solutions within the literature leveraged the knowledge of noise rates to perform loss correction [25, 27, 28], label correction [23, 27], loss reweighting [19], among many other treatments. The importance of knowing the correct noise rate parameters is clearly established. Follow-up works have provided tools for estimating the noise rates without accessing the ground truth label. These include the confident learning approach [26], anchor point [5], and more recently clustering-based ones [36]. Nonetheless, the possibility of unintended harm due to mis-specified noise rates remains. More recent works have looked into robust loss functions that would not require the knowledge of noise rate, hoping to improve the resistance to the mis-specifications [14, 21, 32, 34, 35]. Our work can be viewed as a contribution to this specific line of literature.

Our work also contributes to the understanding of fairness implications when learning with noisy labels and other imperfect information [3, 7, 10, 17, 20, 24, 30, 31]. For instance, [17] considers amending noisy sensitive attributes by appropriately re-scaling the fairness tolerance but is only restricted to class-conditional random noise. [30] proposes two noise-resistant fair classification approaches by constructing unbiased estimators with group-dependent label noise. While most of the existing works would either require the noise rates to be balanced between groups or the knowledge of the noise rates to perform noise correction [30], our work requires neither.

## 2 Preliminaries

We will consider a binary classification problem with $d$-dimensional feature space $X \in \mathbb{R}^d$ and label space $Y \in \{-1, +1\}$. $(X, Y)$ generate according to distribution $\mathcal{D}$. Instead of accessing the clean

data, we consider the setting where the learners only have access to noisy labels $\tilde{Y}$. The generation of $\tilde{Y}$ follows the noise transition model: $e_+(x) := \mathbb{P}(\tilde{Y} = -1|Y = +1, X = x), \ e_-(x) := \mathbb{P}(\tilde{Y} = +1|Y = -1, X = x)$. We further assume $\tilde{Y}$ generates conditionally independently for different $X$. For now, we do not restrict how $e_+(x), e_-(x)$ depend on $x$ but to enable our analysis, later we will. Denote the noisy distribution for $(X, \tilde{Y})$ as $\tilde{\mathcal{D}}$.

In the second constrained learning setting, we assume that for each instance $(X, Y)$, we also observe a sensitive attribute $Z \in \{a, b\}$ that defines two protected groups (e.g., senior v.s. young people) that we will later enforce fairness constraints over.

**Unconstrained learning problem.** In the unconstrained setting, denote the collected training data as $\tilde{D} := \{(x_n, \tilde{y}_n)\}_{n=1}^N$. The goal is to train a classifier $h \in \mathcal{H}$ that minimizes the empirical risk $\mathbb{E}_{\mathcal{D}}[1(h(X) \neq Y)]$ with only accessing $\tilde{D}$, where $1(\cdot)$ is the 0-1 loss function. We will focus on the standard class-dependent noise rate model where $e_+(x) \equiv e_+, e_-(x) \equiv e_-, \forall x$. We assume we have informative labels: $e_+, e_- < 0.5$. Hereby the noise rates depend only on the true label class [25], but not the specific instance $x$ when conditional on the true label. We will operate in the challenging asymmetric error rates setting when $e_+ \neq e_-$.

**Fairness constrained learning problem.** In the fairness constrained setting, denote the collected training data as $\tilde{D} := \{(x_n, \tilde{y}_n, z_n)\}_{n=1}^N$. The goal is to train a classifier $h \in \mathcal{H}$ that maximizes the accuracy while satisfying a certain fairness constraint, measured by a fairness metric $F_z(h)$:

$$\min_{h \in \mathcal{H}} \ \mathbb{E}_{\mathcal{D}}[1(h(X) \neq Y)] \ \ s.t. \ |F_a(h) - F_b(h)| \leq \delta, \tag{1}$$

wherein $\delta > 0$ is a tolerance parameter for fairness violations. Exemplary $F_a(h)$ includes the Equal Opportunity measure $F_a(h) := \mathbb{P}(h(X) = +1|Y = +1, Z = a)$ [12], the demographic parity $F_a(h) := \mathbb{P}(h(X) = +1|Z = a)$ [6], among others.

We assume the error rates are group dependent [30]: $e_+(x) = e_-(x) = e_z, \ z \in \{a, b\}$. Again we assume $e_a, e_b < 0.5$. Within each group, we can further assume that the noise rates are class-dependent, but then we will simply apply our solution within each group (between $+1/-1$ classes) and across the group. We remove this additional layer of complication to stay concise.

**Notations.** We define the following quantities that we will repeatedly use. We will denote by $R_{\mathcal{D}}(h) := \mathbb{E}_{\mathcal{D}}[1(h(X) \neq Y)]$ the true generalization error of a classifier $h$ incurred on the clean distribution $\mathcal{D}$. For a classification-calibrated loss function, we will denote its risk as $R_{\ell, \mathcal{D}}(h) := \mathbb{E}_{\mathcal{D}}[\ell(h(X), Y)]$. Denote the empirical $\ell$-risk of a classifier $h$ on dataset $D$ as $\hat{R}_{\ell, D}(h) := \frac{1}{|D|} \sum_{(x,y) \in D} \ell(h(x), y)$.

Our solution is generically applicable to multi-class settings, which we will explain later and demonstrate with experiments. For a clear exposition of our core idea, we focus on the binary case.

## 3 Equalizing error rates can improve model accuracy

In this section, we focus on the unconstrained learning setting. We will show that while increasing one label class' noise rate raises the generalization errors due to less informative labels, at the same time, it helps remove the training's dependency on the noise rates $(e_+, e_-)$, and therefore improves robustness to possible mis-specification of them. We will quantify such trade-off.

We would mainly demonstrate our idea by comparing to methods that explicitly use the knowledge of noise rates. For one reason, these approaches often have strong theoretical guarantees. The other reason is that the benefit of our approach of not requiring the noise rate would be clearer for this comparison. Particularly, we will use loss correction [25] as the running example. We are aware of the other recent works that do not require the noise rates. First of all, these approaches focus on improving model accuracy, but not the fairness constraints. Secondly, empirically we observe that these approaches can also benefit from our noise rate balancing procedure. We left the discussions to our experiments, where we will demonstrate using one of such losses, peer loss [21].

**Loss correction.** Once knowing $e_+, e_-$, there exist different ways to improve training robustness in the presence of noisy labels. We consider demonstrating our idea using the classical loss correction framework [25, 27]: denote by $\tilde{e}_+, \tilde{e}_-$ the estimated version of $e_+, e_-$. In practice, this knowledge

3

can be obtained by using existing techniques [19, 26, 36]. For a given calibrated loss $\ell$, the loss correction approach defines the following loss:

$$\tilde{\ell}(h(x_n), \tilde{y}_n) := (1 - \tilde{e}_{-sgn(\tilde{y}_n)}) \cdot \ell(h(x_n), \tilde{y}_n) - \tilde{e}_{sgn(\tilde{y}_n)} \cdot \ell(h(x_n), -\tilde{y}_n) \qquad (2)$$

where in above $sgn$ is the sign function. A favorable property of $\tilde{\ell}$ is its unbiasedness (expectation over $\tilde{Y}|Y = y$) w.r.t. $\ell(h(x), y)$ when given the correct noise rate parameters $e_+, e_-$. For readers who need more background knowledge of loss correction, we reproduce details in Appendix **??**. A classifier will be trained following empirical risk minimization (ERM) using $\tilde{\ell}$: $h^*_{\tilde{\ell},\tilde{D}} := \arg\min_{h \in \mathcal{H}} \sum_{n=1}^N \tilde{\ell}(h(x_n), \tilde{y}_n)$. Compared to the standard loss correction, we omit the $1 - \tilde{e}_- - \tilde{e}_+$ term from the denominator, which will not affect the minimizer of our ERM problem. Denote $h^*_{\ell,\mathcal{D}} = \arg\min_{h \in \mathcal{H}} R_{\ell,\mathcal{D}}(h)$, the optimal classifier for $R_{\ell,\mathcal{D}}(h)$, and $\mathrm{err_M} := \max\{|\tilde{e}_+ - e_+|, |\tilde{e}_- - e_-|\}$ the maximal mis-specification error. When $\ell$ is $L$-Lipschitz and bounded by $\bar{\ell}$, let $L_1 := 4L, L_2 := 2\bar{\ell} > 0$, adapting the proof from [25] we show:

**Theorem 1.** *For any $\delta > 0$, we have with probability at least $1 - \delta$:*

$$R_{\ell,\mathcal{D}}(h^*_{\tilde{\ell},\tilde{D}}) - R_{\ell,\mathcal{D}}(h^*_{\ell,\mathcal{D}}) \leq \underbrace{\frac{1}{1 - e_+ - e_-}}_{\text{label informativeness}} \cdot L_1 \cdot \mathcal{R}(\mathcal{H}) + \underbrace{\frac{\mathrm{err_M}}{1 - e_+ - e_-}}_{\text{mis-specification}} \cdot L_2 + 2\sqrt{\frac{\log 1/\delta}{2N}}, \quad (3)$$

*where in above $\mathcal{R}(\mathcal{H})$ denotes the Rademacher complexity of $\mathcal{H}$.*

Clearly, $e_+, e_-$ control the error due to label informativeness: a pair of $e_+, e_-$ with higher sum $(e_+ + e_-)$ induce larger generalization error bound. On the other hand, the potential imperfect estimates of them introduce additional error due to the mis-specification.

### 3.1 When does equalizing error rate improve performance?

Now we first provide intuitions for why increasing noise rate to balance $e_+$ and $e_-$ might be considered helpful. Without loss of generality, suppose $e_+ > e_-$. When we increase $e_-$ to match $e_+$ such that $e = e_+ = e_-$ (later we will explain how we do so), denote by $\hat{Y}$ the newly generated noisy label with $e$ symmetric error rates for both classes. Note that $e < 0.5$. Denote by $\hat{y}_n$ the generated label for example $n$, $\hat{\mathcal{D}}$ the distribution of $(X, \hat{Y})$, and $\hat{D}$ the dataset $\{(x_n, \hat{y}_n)\}_{n=1}^N$. We first show:

**Lemma 2.** *When $e < 0.5$, minimizing $\mathbb{P}(h(X) \neq \hat{Y})$ is equivalent with minimizing $\mathbb{P}(h(X) \neq Y)$.*

The above lemma says that the minimizer of $\mathbb{P}(h(X) \neq \hat{Y})$ is equivalent to the optimal classifier for $\mathbb{P}(h(X) \neq Y)$ when the hypothesis space $\mathcal{H}$ covers the optimal $h$! Therefore when $\ell$ is classification-calibrated on the noisy distribution $\hat{\mathcal{D}}$, and when the hypothesis space is sufficiently large, we know that the optimal classifier for $\mathbb{E}_{\hat{\mathcal{D}}}[\ell(h(X), \hat{Y})]$ will be exactly the same as the one for $\mathbb{E}_{\mathcal{D}}[\ell(h(X), Y)]$. This further implies that when the error rates are symmetric, the training might not need the error rates information, and performing ERM on the noisy data:

$$h^*_{\ell,\hat{D}} := \arg\min_{h \in \mathcal{H}} \sum_{n=1}^N \ell(h(x_n), \hat{y}_n) \qquad (4)$$

suffices to find the optimal classifier for the clean distribution, when given a sufficient amount of data. The above argument is slightly trickier when considering a finite hypothesis space $\mathcal{H}$. Denote by $-h$ the classifier that always flips the prediction from $h$. Then we prove that

**Theorem 3.** *Suppose 1) $\ell$ is symmetric s.t. $\ell(h(x), -y) = \ell(-h(x), y)$ and 2) $-h^*_{\ell,\hat{D}}, -h^*_{\ell,\mathcal{D}} \in \mathcal{H}$: $\hat{R}_{\ell,\hat{D}}(-h^*_{\ell,\hat{D}}) \geq \hat{R}_{\ell,\hat{D}}(-h^*_{\ell,\mathcal{D}})$. Then for any $\delta > 0$, we have with probability at least $1 - \delta$:*
$R_{\ell,\mathcal{D}}(h^*_{\ell,\hat{D}}) - R_{\ell,\mathcal{D}}(h^*_{\ell,\mathcal{D}}) \leq \frac{1}{1-2e} \cdot L_1 \cdot \mathcal{R}(\mathcal{H}) + 2\sqrt{\frac{\log 1/\delta}{2N}}.$

The second condition above is simply stating that the opposite classifier of the empirically optimal one incurs a high empirical loss, and particularly higher than $-h^*_{\ell,\mathcal{D}}$. Intuitively, this condition is satisfied for binary classification — if one classifier performs the best on the empirical data, flipping its prediction often results in a very wrong one. Now comparing Theorem 1 and 3 we observe that equalizing noise rates increases the error due to loss of informative labels by:

$$\left| \frac{1}{1-2e} - \frac{1}{1-e_+-e_-} \right| \cdot L_1 \cdot \mathcal{R}(\mathcal{H}) = \frac{|e_+ - e_-|}{|1-e_+-e_-| \cdot |1-2e_+|} \cdot L_1 \cdot \mathcal{R}(\mathcal{H}).$$

On the other hand, equalizing noise rates avoids the error due to mis-specifying $e_+, e_-$ by $\frac{\text{err}_M}{1-e_+-e_-} \cdot L_2$. Therefore, it is considered helpful to increase $e_-$ to $e_+$ when $\text{err}_M \geq \frac{|e_+-e_-| \cdot L_1 \cdot \mathcal{R}(\mathcal{H})}{|1-2e_+| \cdot L_2}$. The above condition is likely to hold when the gap between noise rates $|e_+ - e_-|$ is sufficiently small, and when we do not have high confidence in the estimation of $e_+, e_-$ (therefore a high $\text{err}_M$).

## 4  Equalizing error rates improves fairness guarantee

In this section, we consider the fairness constrained learning problem and show that equalizing noise rates helps improve fairness guarantee when only accessing the noisy labels. There are generally two types of fairness constraints: those that do not depend on the label and those that do. For the ones that do not, for example, demographic parity $\mathbb{P}(h(X) = +1|Z = a) = \mathbb{P}(h(X) = +1|Z = b)$, the existence of noisy labels does not impose additional challenges when enforcing such constraints. For those that do, typically they are functions of true positive rates (TPR, also known as the equal opportunity measure) and false positive rates (FPR). It was shown in [30] that equalizing TPR and FPR on the noisy label leads to disparities when $e_a \neq e_b$. We will reproduce the same observations. We focus on the second type of constraints: equalizing TPR and FPR using the noisy labels.

Having noted that equalizing TPR and FPR on the noisy labels leads to unintended consequences, the literature has observed recent works on performing fairness constraint correction [30]. Similar to the loss correction approaches, such correction methods would again require the knowledge of $e_a, e_b$, which is subject to mis-specification error. Define

$$\text{TPR}_z(h) := \mathbb{P}(h(X) = +1|Y = +1, Z = z), \ \text{FPR}_z(h) := \mathbb{P}(h(X) = +1|Y = -1, Z = z)$$
$$\widetilde{\text{TPR}}_z(h) := \mathbb{P}(h(X) = +1|\tilde{Y} = +1, Z = z), \ \widetilde{\text{FPR}}_z(h) := \mathbb{P}(h(X) = +1|\tilde{Y} = -1, Z = z)$$

Resampling the noisy data examples such that $\mathbb{P}(\tilde{Y} = +1|Z = z) = \mathbb{P}(\tilde{Y} = -1|Z = z) = 0.5, z \in \{a, b\}$, define $C_{z,1} := 0.5 \cdot e_z, C_{z,2} := 0.5 \cdot (1 - e_z)$, we derive the following relationship:

**Lemma 4.** *$TPR_z(h)$, $FPR_z(h)$ relate to $\widetilde{TPR}_z(h)$, $\widetilde{FPR}_z(h)$ as follows:*

$$TPR_z(h) = \frac{C_{z,1} \cdot \widetilde{TPR}_z(h) - C_{z,2} \cdot \widetilde{FPR}_z(h)}{e_z - 0.5}, \ \ FPR_z(h) = \frac{C_{z,1} \cdot \widetilde{FPR}_z(h) - C_{z,2} \cdot \widetilde{TPR}_z(h)}{e_z - 0.5} \tag{5}$$

The above lemma first implies that since two groups $a, b$ might have different $C_{z,1}, C_{z,2}$, equalizing $\widetilde{\text{TPR}}, \widetilde{\text{FPR}}$ using the noisy labels naively is insufficient to guarantee equalization of TPR, FPR.

**Unfairness due to model error.**  On the other hand, Lemma 4 points out a way to perform constraint correction using the knowledge of $e_a, e_b$. Denote by $\tilde{e}_a, \tilde{e}_b$ (both $< 0.5$) the estimated copies of $e_a, e_b$ that we have access to. Suppose we suffer from the following mis-specifications: $\text{err}_M := \min\{\text{err}_a := |\tilde{e}_a - e_a|, \text{err}_b := |\tilde{e}_b - e_b|\}$. Denote the corrected TPR and FPR using $\widetilde{\text{TPR}}$ and $\widetilde{\text{FPR}}$ as well as $\tilde{e}_a, \tilde{e}_b$ as $\text{TPR}_z^c(h), \text{FPR}_z^c(h)$: Define $\tilde{C}_{z,1} := 0.5 \cdot \tilde{e}_z, \tilde{C}_{z,2} := 0.5 \cdot (1 - \tilde{e}_z)$, and:

$$\text{TPR}_z^c(h) = \frac{\tilde{C}_{z,1} \cdot \widetilde{\text{TPR}}_z(h) - \tilde{C}_{z,2} \cdot \widetilde{\text{FPR}}_z(h)}{\tilde{e}_z - 0.5}, \ \ \text{FPR}_z^c(h) = \frac{\tilde{C}_{z,1} \cdot \widetilde{\text{FPR}}_z(h) - \tilde{C}_{z,2} \cdot \widetilde{\text{TPR}}_z(h)}{\tilde{e}_z - 0.5} \tag{6}$$

Theorem 5 establishes possible fairness violation due to $\text{err}_M$, noise rates mis-specification:

**Theorem 5.** *Equalizing $TPR_z^c(h)$ & $FPR_z^c(h)$ for $a, b$ leads to following possible fairness violation:*

$$|TPR_a(h) - TPR_b(h)| \geq \text{err}_M \cdot \left| \frac{\widetilde{TPR}_a(h)}{(2e_a - 1)(2\tilde{e}_a - 1)} - \frac{\text{err}_b}{\text{err}_a} \frac{\widetilde{TPR}_b(h)}{(2e_b - 1)(2\tilde{e}_b - 1)} \right|,$$

$$|FPR_a(h) - FPR_b(h)| \geq \text{err}_M \cdot \left| \frac{\widetilde{FPR}_a(h)}{(2e_a - 1)(2\tilde{e}_a - 1)} - \frac{\text{err}_b}{\text{err}_a} \frac{\widetilde{FPR}_b(h)}{(2e_b - 1)(2\tilde{e}_b - 1)} \right|.$$

But as a consequence of Lemma 4, we immediately know that

**Theorem 6.** *When $e_a = e_b$, equalizing $\widetilde{TPR}$ and $\widetilde{FPR}$ suffices to equalizing the true TPR and FPR.*

This is simply because equalizing $e_a$ and $e_b$ also equalizes both $C_{z,1}$ and $C_{z,2}$. Theorem 6 helps us establish a strong fairness guarantee: as long as we know the training data has balanced label noise rates across different groups, enforcing the constraints directly on the noisy data suffices to guarantee the equality between true TPR and FPR.

Our above observation points out increasing one group's label noise to match another helps establish the fairness guarantees more easily. Consider the case $e_a > e_b$. Now suppose we are able to increase $e_b$ to match $e_a$ such that $e = e_a = e_b$. Denote by $\hat{Y}$ the newly generated noisy label with $e$ symmetric error rates for both groups, and correspondingly $\hat{y}_n$ the newly generated training labels. Then performing fairness constrained ERM directly on $\hat{D} := \{(x_n, \hat{y}_n, z_n)\}_n$:

$$h^*_{\ell, \hat{D}} := \arg\min_{h \in \mathcal{H}} \sum_{n=1}^{N} \ell(h(x_n), \hat{y}_n) \ \ s.t. \ |\hat{F}_a(h) - \hat{F}_b(h)| \leq \delta, \tag{7}$$

will help us equalize TPR and FPR between two groups when the number of training data is sufficiently large. In above, $\hat{F}_a(h), \hat{F}_b(h)$ are the empirically computed fairness measures using $\hat{D}$.

## 5 Identifying cleaner class without using ground truth label

With the aforementioned benefits of balancing the noise rates, the remaining technical question is to determine how does $e_+$ or $e_a$ compare to $e_-$ or $e_b$? Of course, when we have access to ground truth labels, we will have a rough estimate and carry on to insert label noise (e.g., by further randomly flipping the noisy labels, which we will discuss at the end of this section). In this section, we describe another approach without the need for ground truth labels. We prove its sufficiency in identifying the noisier class of labels. We demonstrate our idea using the unconstrained learning setting for detecting $sgn(e_+ - e_-)$, but the idea easily generalizes to the fairness setting to detect the order of $e_a$ and $e_b$.

### 5.1 Noisy label agreements

We first present the following definition:

**Definition 7** (Clusterability). *We say the dataset $D$ satisfies 2-NN clusterability if each instance $x$ shares the same true label class with its two nearest neighbors measured by $||x - x'||_2$.*

For the rest of the section, we will assume that $D$ satisfies 2-NN clusterability. 2-NN was similarly introduced in a recent work [36] and has been shown to be a requirement that is mild to satisfy. Now we define the following two quantities that are central to the development of our idea. For an arbitrary instance $X_1$ with noisy label $\tilde{Y}_1$, denote the noisy labels for two nearest neighbor instances of $X_1$ as $\tilde{Y}_2, \tilde{Y}_3$. Define the following agreement measures:

**Definition 8** (2-NN Agreements). *Let $\tilde{Y}_1$ denote the noisy label for a randomly selected instance $X_1$. $\tilde{Y}_2, \tilde{Y}_3$ are the noisy labels of $X_1$'s 2-NN instances (measured by $||x - x'||_2$).*

$$\text{Positive Agreements} \quad PA_{\mathcal{D}} := \mathbb{P}(\tilde{Y}_2 = \tilde{Y}_3 = +1 | \tilde{Y}_1 = +1) \tag{8}$$

$$\text{Negative Agreements} \quad NA_{\mathcal{D}} := \mathbb{P}(\tilde{Y}_2 = \tilde{Y}_3 = -1 | \tilde{Y}_1 = -1) \tag{9}$$

$PA_{\mathcal{D}}$ computes the likelihood of the neighbor data points "agreeing" on a positive label. $NA_{\mathcal{D}}$ computes the one for the negative label. Now we will first sub-sample the noisy distribution and compute $PA_{\mathcal{D}}, NA_{\mathcal{D}}$:

- **Step 1** Sample $\tilde{Y}$ such that $\mathbb{P}(\tilde{Y} = +1) = \mathbb{P}(\tilde{Y} = -1) = 0.5$. Denote this resampled distribution as $\mathcal{D}^\diamond$.
- **Step 2** Compute $NA_{\mathcal{D}^\diamond}, PA_{\mathcal{D}^\diamond}$.

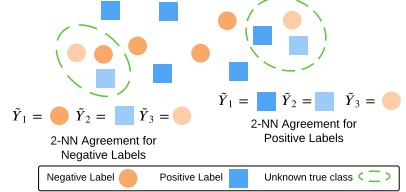

Next we prove that knowing $NA_{\mathcal{D}^\diamond}, PA_{\mathcal{D}^\diamond}$ suffices to inform the order between $e_+, e_-$ ($sgn(e_+ - e_-)$):

Figure 1: **2-NN Agreements.** "Transparent" ones are the solid instance's 2-NN instances.

**Theorem 9.** *When $e_+, e_- < 0.5$ and $D$ satisfies 2-NN clusterability, $PA_{\mathcal{D}^\diamond}, NA_{\mathcal{D}^\diamond}$ relate to $e_+, e_-$ as follows: $PA_{\mathcal{D}^\diamond} - NA_{\mathcal{D}^\diamond} = 2(0.5 - e_+)(0.5 - e_-)(e_- - e_+)$. Then if $PA_{\mathcal{D}^\diamond} > NA_{\mathcal{D}^\diamond}$, we know that $e_+ < e_-$; otherwise $e_+ > e_-$. If $PA_{\mathcal{D}^\diamond} = NA_{\mathcal{D}^\diamond}$, then $e_+ = e_-$.*

**Estimating PA$_{\mathcal{D}^\diamond}$, NA$_{\mathcal{D}^\diamond}$ using $\mathcal{D}^\diamond$.** Note that both PA$_{\mathcal{D}^\diamond}$ and NA$_{\mathcal{D}^\diamond}$, $\mathbb{P}(\tilde{Y}_2 = \tilde{Y}_3 = +1|\tilde{Y}_1 = +1)$ and $\mathbb{P}(\tilde{Y}_2 = \tilde{Y}_3 = -1|\tilde{Y}_1 = -1)$ can be counted from the noisy data without accessing the ground truth. For instance for PA$_{\mathcal{D}^\diamond}$:

- **Step 1** Find all $x_n$ in $\mathcal{D}^\diamond$ with $\tilde{y}_n = +1$ and their 2-NN $n_1, n_2$.
- **Step 2** Count $\frac{\#\{n:\ \tilde{y}_n = \tilde{y}_{n_1} = \tilde{y}_{n_2} = +1\}}{\#\{n:\ \tilde{y}_n = +1\}}$. $\#(\cdot)$ counts the number of samples that satisfy the condition.

We show pseudocode for an implementation of estimating PA in Figure **??** in Appendix **??**.

## 5.2 Our solution: NOISE+

We show that randomly flipping $\tilde{Y} = y$ with the smaller $e_{sgn(y)}$ by a small probability $\epsilon$ monotonically decreases the gap between noise rates $|e_+ - e_-|$. Without the loss of generality suppose $e_+ < e_-$, and we will only flip the $\tilde{Y} = +1$ labels (but not flipping the ones with $\tilde{Y} = -1$). We show numpy-like pseudocode for the flipping function in Figure 3, and the implementation for exclusively flipping $\tilde{Y} = -1$ labels is symmetric. Denote by $\hat{Y}$ as the flipped version of $\tilde{Y}$: $\mathbb{P}(\hat{Y} = -1|\tilde{Y} = +1) = \epsilon$, and $\hat{e}_+ := \mathbb{P}(\hat{Y} = -1|Y = +1), \hat{e}_- := \mathbb{P}(\hat{Y} = +1|Y = -1)$. We have:

**Proposition 10.** $\hat{e}_+ = (1 - e_+) \cdot \epsilon + e_+, \ \hat{e}_- = (1 - \epsilon) \cdot e_-$. *Further, the new gap between the noise rates of the flipped label $\hat{Y}$ is a monotonic function of $\epsilon$: $\hat{e}_- - \hat{e}_+ = e_- - e_+ - (1 - e_+ + e_-) \cdot \epsilon$.*

Since $1 - e_+ + e_- > 0$, when $\epsilon$ is small, the above derivation shows the effectiveness in reducing the noise rate gap $e_- - e_+$ by randomly flipping the noisy labels that correspond to the class with lower noise rate. The only remaining question is how to find the optimal $\epsilon$ s.t. $\hat{e}_- - \hat{e}_+ = 0$. Calling Theorem 9, we know PA$_{\mathcal{D}^\diamond}$ $-$ NA$_{\mathcal{D}^\diamond}$ $= 2(0.5 - \hat{e}_+)(0.5 - \hat{e}_-)(\hat{e}_- - \hat{e}_+)$. Denote by $f(\epsilon) := 0.5 \cdot ($PA$_{\mathcal{D}^\diamond}$ $-$ NA$_{\mathcal{D}^\diamond})$.

Easy to derive the three solutions for $f(\epsilon) = 0$ (setting each of the terms to 0): $\epsilon_1 = 1 - \frac{0.5}{e_-} < 0$, $\epsilon_2 = \frac{e_- - e_+}{1 - e_+ + e_-}$, $\epsilon_3 = \frac{0.5 - e_+}{1 - e_+}$: note that $\epsilon_2 = \frac{e_- - e_+}{1 - e_+ + e_-} < \frac{(e_- - e_+) + (1 - e_+ - e_-)}{(1 - e_+ + e_-) + (1 - e_+ - e_-)} = \frac{1 - 2e_+}{2(1 - e_+)} = \epsilon_3$, and $\epsilon_3$ will lead to an uninformative state where $\hat{e}_+ = 0.5$. Therefore $\epsilon_2$ is our target root.

The monotonicity of $f(\epsilon)$ from 0 to $\epsilon_2$ motivates us to look for a proper $\epsilon$ by a binary search procedure. Suppose we have two different flipping parameters $\epsilon_l < \epsilon_r$. Initially the synthetic datasets induced by them are $\mathcal{D}_l$ and $\mathcal{D}_r$, and the gaps of the counted agreements are $C_l = $ PA$_{\mathcal{D}_l}$ $-$ NA$_{\mathcal{D}_l}$ and $C_r = $ PA$_{\mathcal{D}_r}$ $-$ NA$_{\mathcal{D}_r}$, satisfying $C_l > 0 > C_r$. In each iteration, we try a new flip parameter $\epsilon_{\text{mid}} = (\epsilon_l + \epsilon_r)/2$, and check whether the new gap $C_{\text{mid}}$ is bounded by a threshold $\gamma$. If $-\gamma \leq C_{\text{mid}} \leq \gamma$, we return the labels flipped by $\epsilon_{\text{mid}}$. Otherwise, we update the values of $\epsilon_l$ and $\epsilon_r$ according to the sign of $C_{\text{mid}}$: if $C_{\text{mid}} < 0$, we set $\epsilon_r \leftarrow \epsilon_{\text{mid}}, \mathcal{D}_r \leftarrow \mathcal{D}_{\text{mid}}$ (reducing $\epsilon_r$); otherwise $\epsilon_l \leftarrow \epsilon_{\text{mid}}, \mathcal{D}_l \leftarrow \mathcal{D}_{\text{mid}}$ (increasing $\epsilon_l$). We summarize NOISE+ in Algorithm 1. Here, we initialize $\epsilon_r = 0.3$ which empirically succeeds in varied noise settings. In case $\epsilon_r = 0.3$ fails, we can grid search a variety of $\epsilon$ values (e.g., 0.1, 0.2) satisfying $C_r < 0$ as the initial $\epsilon_r$ to proceed our binary search algorithm. Note that Algorithm 1 assumes $e_+ < e_-$, and the implementation is symmetric for $e_+ > e_-$.

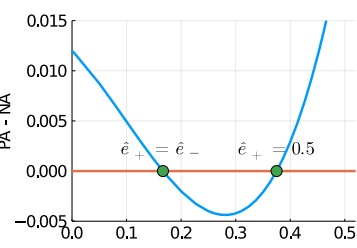

Figure 2: **Agreement gap PA$-$NA varies for different $\epsilon$.** There are only two positive roots for PA $-$ NA $= 0$. The less one results in $\hat{e}_+ = \hat{e}_-$.

**Selection of loss function.** Our algorithm is particularly suitable for losses that do not require the knowledge of noise rates. Standard cross entropy (CE) would certainly be applicable. Other robust loss functions are great options too. We note a recently proposed loss, peer loss function [21], does not require the specification of noise rates:

$$\ell_{\text{peer}}(h(x_n), \tilde{y}_n) := \ell\left(h(x_n), \tilde{y}_n\right) - \alpha \cdot \ell\left(h(x_{p_1}), \tilde{y}_{p_2}\right).$$

Here, $\alpha > 0$ is a hyper-parameter to control the balance of the instances for each label, $p_1$ and $p_2$ are uniformly and randomly selected peer samples. Peer loss is also promoted in [30] for the fairness constrained setting. We believe it is particularly suitable for our NOISE+. We will evaluate CE and peer loss in the experiment section.

**Fairness constrained learning.** Our above derivations and observations can be similarly applied to the fairness constrained setting. The only difference lies in that now $\text{PA}_{\mathcal{D}^\diamond}, \text{NA}_{\mathcal{D}^\diamond}$ are corresponding to the agreements within each of the two groups $a, b$ instead of the two label class group $\tilde{Y} = +1$ and $\tilde{Y} = -1$. We provide details in the Appendix **??** without repeating them here.

**Training.** Once we have the output and noise-incremented dataset $\hat{D} := \{(x_n, \hat{y}_n)\}_n$ from Algorithm 1, we will use it directly in our ERM framework for the unconstrained case as in Eqn. (4) and will use it in the fairness constrained ERM in Eqn. (7) directly too.

**Extension to multi-class.** As explained at the beginning, our algorithm can largely extend to the multi-class/group setting. The primary requirement of the extension is to extend the definition of $\text{PA}_{\mathcal{D}^\diamond}, \text{NA}_{\mathcal{D}^\diamond}$ to each label class/group. We defer details to Appendix **??**. In the next section, we will demonstrate the effectiveness of our results with the CIFAR-10 dataset.

---

**Algorithm 1** NOISE+: A binary search algorithm for balancing noise rates.

---

**Require:** $\gamma > 0, \epsilon_l = 0, \epsilon_r = 0.3$
  Resample a balanced set $\mathcal{D}^\diamond$ from $\tilde{D}$;
  Initialize $\mathcal{D}_l = \mathcal{D}^\diamond, \mathcal{D}_r = \text{Flip}(\mathcal{D}^\diamond, \epsilon_r)$;
  Estimate $\text{PA}_{\mathcal{D}_l}, \text{PA}_{\mathcal{D}_r}, \text{NA}_{\mathcal{D}_l}, \text{NA}_{\mathcal{D}_r}$.
  **while** $\text{PA}_{\mathcal{D}_l} - \text{NA}_{\mathcal{D}_l} > \gamma$ and $\text{PA}_{\mathcal{D}_r} - \text{NA}_{\mathcal{D}_r} < -\gamma$ **do**
    $\epsilon_{\text{mid}} \leftarrow (\epsilon_l + \epsilon_r)/2, \mathcal{D}_{\text{mid}} \leftarrow \text{Flip}(\mathcal{D}^\diamond, \epsilon_{\text{mid}})$;
    Estimate $\text{PA}_{\mathcal{D}_{\text{mid}}}$ and $\text{NA}_{\mathcal{D}_{\text{mid}}}$;
    **if** $\text{PA}_{\mathcal{D}_{\text{mid}}} - \text{NA}_{\mathcal{D}_{\text{mid}}} < -\gamma$ **then**
      /* $\epsilon_{\text{mid}}$ is at the right of the root */
      $\epsilon_r \leftarrow \epsilon_{\text{mid}}, \mathcal{D}_r \leftarrow \mathcal{D}_{\text{mid}}$;
      $\text{PA}_{\mathcal{D}_r} \leftarrow \text{PA}_{\mathcal{D}_{\text{mid}}}, \text{NA}_{\mathcal{D}_r} \leftarrow \text{NA}_{\mathcal{D}_{\text{mid}}}$;
    **else if** $\text{PA}_{\mathcal{D}_{\text{mid}}} - \text{NA}_{\mathcal{D}_{\text{mid}}} > \gamma$ **then**
      /* $\epsilon_{\text{mid}}$ is at the left of the root */
      $\epsilon_l \leftarrow \epsilon_{\text{mid}}, \mathcal{D}_l \leftarrow \mathcal{D}_{\text{mid}}$;
      $\text{PA}_{\mathcal{D}_l} \leftarrow \text{PA}_{\mathcal{D}_{\text{mid}}}, \text{NA}_{\mathcal{D}_l} \leftarrow \text{NA}_{\mathcal{D}_{\text{mid}}}$;
    **else**
      **return** $\hat{D} = \text{Flip}(\tilde{D}, (\epsilon_l + \epsilon_r)/2)$;
    **end if**
  **end while**
  **return** unsuccessful;

```python
def flip(dataset, epsilon):
    # unpack the dataset
    (X, y) = dataset
    # filter positives
    y_plus = y[y > 0]
    # flip the positive
        labels w.p. epsilon
    is_flipped = numpy.
        random.binomial(1,
        epsilon, y_plus.size
        )
    y[y > 0] = is_flipped *
        (-y_plus) + (1 -
        is_flipped) * y_plus
    # pack and return the
        dataset
    return (X, y)
```

Figure 3: **Pseudocode for** Flip. Flip takes the dataset and a small probability $\epsilon$ as input, and only flips positive examples with probability $\epsilon$.

## 6 Experiments

In order to verify the power of our increasing-to-balancing method, we conduct extensive experiments on both unconstrained learning and constrained learning settings. The datasets include: the UCI Adult Income dataset [9], the Compas recidivism dataset [2], Fairface [15] face attribute dataset, and CIFAR 10 [16] dataset. We defer more dataset details to Appendix **??**.

### 6.1 Unconstrained learning

**Setup.** For the unconstrained learning, we implement a one-layer perceptron for binary classification on Adult, Compas, and Fairface datasets. We flip the true labels on training set according to a set of asymmetric noise rates, train the models on the corrupted labels, and evaluate the accuracy on test set with clean labels. The baseline methods we compare include: surrogate loss [25] with mis-specified noise rates (**Mis. SL**) and estimated noise rates (**Est. SL**) respectively, vanilla cross entropy (**CE**), and peer loss functions [21] (**Peer**). For Mis. SL, we randomly generate the noise parameters but ensure that the trace is equal to that of the true noise transition matrix. For Est. SL, we estimate the noise rates by adopting the confident learning procedure [26]. We note that the training of peer loss functions does not require the knowledge of noise rates, and follow their instructions to tune the hyper-parameters through cross-validation. For the same corrupted training labels, we deploy our increasing-to-balancing procedure, and evaluate the performance of cross entropy and peer loss functions again. We set the threshold $\gamma$ used in Algorithm 1 as $0.1\%$ on Adult and Fairface datasets, and loose it to $1\%$ on Compas due to its much smaller data size. We repeat all the methods 5 runs with different random seeds and report the mean and standard deviation.

Table 1: **Binary classification accuracy of compared methods on 3 datasets across different levels of noise rates.** Mis. SL: surrogate loss [25] with misspecified parameters. Est. SL: surrogate loss [25] with estimated parameters. CE: vanilla cross entropy. Peer: peer loss function [21]. All methods are trained with one-layer perceptron with the same hyper-parameters. For each noise setting, we average across 5 runs and report the mean and standard deviation. We find that the increasing-to-balancing can boost the vanilla cross entropy on all the noise settings. We highlight any boost of increasing-to-balancing for vanilla CE in green, and the best accuracy in blue.

| Dataset | $e_-$ | $e_+$ | BASELINES (LESS NOISE) | | | | NOISE+ (MORE NOISE) | |
| | | | Mis. SL | Est. SL | CE | Peer | CE | Peer |
|---|---|---|---|---|---|---|---|---|
| Adult | 0.0 | 0.1 | $72.79 \pm 0.34$ | $72.64 \pm 0.38$ | $72.63 \pm 0.29$ | $72.77 \pm 0.32$ | $73.62 \pm 0.37$ | $73.86 \pm 0.41$ |
| $n = 48,842$ | 0.0 | 0.2 | $72.27 \pm 0.39$ | $72.13 \pm 0.37$ | $71.26 \pm 0.38$ | $71.95 \pm 0.34$ | $72.73 \pm 0.71$ | $73.52 \pm 0.58$ |
| $d = 28$ | 0.1 | 0.2 | $73.02 \pm 0.50$ | $72.68 \pm 0.16$ | $72.31 \pm 0.25$ | $72.88 \pm 0.14$ | $71.92 \pm 1.98$ | $73.81 \pm 0.40$ |
| | 0.1 | 0.3 | $72.44 \pm 0.47$ | $72.15 \pm 0.43$ | $69.06 \pm 2.01$ | $72.26 \pm 0.43$ | $69.53 \pm 4.90$ | $73.34 \pm 1.27$ |
| | 0.2 | 0.3 | $72.81 \pm 0.51$ | $72.43 \pm 0.14$ | $71.44 \pm 0.93$ | $72.78 \pm 0.28$ | $71.55 \pm 2.04$ | $73.75 \pm 0.26$ |
| | 0.2 | 0.4 | $72.06 \pm 0.19$ | $71.97 \pm 0.41$ | $63.49 \pm 1.58$ | $71.97 \pm 0.37$ | $65.99 \pm 7.99$ | $71.43 \pm 2.26$ |
| | 0.3 | 0.4 | $52.65 \pm 0.53$ | $72.67 \pm 0.26$ | $71.55 \pm 0.88$ | $73.49 \pm 0.18$ | $72.54 \pm 1.84$ | $74.27 \pm 0.20$ |
| Compas | 0.0 | 0.1 | $66.36 \pm 1.05$ | $66.04 \pm 1.14$ | $66.16 \pm 1.13$ | $68.06 \pm 0.70$ | $67.14 \pm 0.92$ | $68.22 \pm 0.68$ |
| $n = 7,168$ | 0.0 | 0.2 | $66.84 \pm 0.69$ | $66.06 \pm 0.81$ | $65.38 \pm 1.40$ | $68.03 \pm 0.77$ | $66.51 \pm 1.90$ | $68.40 \pm 0.78$ |
| $d = 10$ | 0.1 | 0.2 | $66.41 \pm 0.43$ | $65.69 \pm 0.57$ | $65.91 \pm 0.97$ | $67.49 \pm 0.40$ | $66.54 \pm 0.21$ | $67.80 \pm 0.44$ |
| | 0.1 | 0.3 | $65.91 \pm 0.42$ | $65.22 \pm 0.63$ | $61.24 \pm 0.70$ | $67.36 \pm 0.79$ | $65.76 \pm 2.09$ | $68.05 \pm 0.56$ |
| | 0.2 | 0.3 | $65.06 \pm 0.72$ | $65.86 \pm 1.69$ | $65.06 \pm 1.48$ | $68.02 \pm 0.94$ | $66.46 \pm 1.27$ | $68.04 \pm 1.11$ |
| | 0.2 | 0.4 | $64.82 \pm 0.52$ | $65.47 \pm 0.46$ | $59.68 \pm 2.49$ | $67.37 \pm 0.54$ | $63.85 \pm 3.31$ | $68.39 \pm 0.56$ |
| Fairface | 0.0 | 0.1 | $87.64 \pm 0.03$ | $87.75 \pm 0.03$ | $87.41 \pm 0.11$ | $87.58 \pm 0.15$ | $88.23 \pm 0.07$ | $88.49 \pm 0.12$ |
| $n = 108,501$ | 0.0 | 0.2 | $85.22 \pm 0.06$ | $85.83 \pm 0.08$ | $85.08 \pm 0.16$ | $85.18 \pm 0.16$ | $88.55 \pm 0.03$ | $88.67 \pm 0.03$ |
| $d = 50$ | 0.1 | 0.2 | $87.67 \pm 0.07$ | $87.56 \pm 0.04$ | $87.21 \pm 0.08$ | $87.28 \pm 0.05$ | $88.45 \pm 0.06$ | $88.65 \pm 0.07$ |
| | 0.1 | 0.3 | $72.03 \pm 0.13$ | $85.68 \pm 0.07$ | $83.20 \pm 0.12$ | $84.58 \pm 0.09$ | $87.81 \pm 0.14$ | $88.50 \pm 0.12$ |
| | 0.2 | 0.3 | $74.18 \pm 0.20$ | $87.34 \pm 0.14$ | $86.47 \pm 0.09$ | $87.00 \pm 0.11$ | $88.46 \pm 0.08$ | $88.58 \pm 0.10$ |
| | 0.2 | 0.4 | $58.30 \pm 0.23$ | $85.48 \pm 0.09$ | $78.33 \pm 0.63$ | $84.05 \pm 0.13$ | $81.90 \pm 0.58$ | $87.69 \pm 0.15$ |

**Results.** We show the experimental results in Table 1. We observe that our increasing-to-balancing method significantly improves the accuracy of the vanilla cross entropy on the majority of noise settings. It is notable that the cross entropy with increasing-to-balancing has a comparable performance with Est. SL, when the noise is relatively small. When the noise rates are large (e.g. 0.2 and 0.4), we observe a significant degradation for cross entropy, but increasing-to-balancing still boosts the accuracy. Moreover, we find that peer loss after increasing-to-balancing is more robust to noise, and dominantly achieve the highest accuracy.

Table 2: **Accuracy of compared methods across different levels of noise gap for multi-class classification.**

| Dataset | noise gap | LESS NOISE | | | | MORE NOISE | |
| | | Mis. SL | Est. SL | CE | Peer | CE | Peer |
|---|---|---|---|---|---|---|---|
| MNIST | 0.1 | 89.59 | 89.69 | 86.66 | 88.12 | 86.81 | 89.19 |
| | 0.2 | 88.10 | 88.61 | 84.53 | 87.21 | 85.97 | 89.12 |
| | 0.3 | 84.97 | 86.88 | 85.24 | 86.35 | 81.89 | 88.75 |
| CIFAR-10 | 0.1 | 70.90 | 85.76 | 88.03 | 89.66 | 88.69 | 89.90 |
| | 0.2 | 80.51 | 86.34 | 88.43 | 89.36 | 89.01 | 90.08 |
| | 0.3 | 81.30 | 90.61 | 89.78 | 90.24 | 87.98 | 89.92 |

Table 3: **Accuracy of compared methods on Clothing1M dataset.**

| Method | Test Accuracy |
|---|---|
| CE | 68.94% |
| Loss Correction [27] | 69.84% |
| Co-Teaching [11] | 70.15% |
| CE + NOISE+ | **70.37%** |

**Multi-class extension.** We test NOISE+ in the multi-class setting by balancing the noise rates class by class. We evaluate the compared methods on MNIST [18] and CIFAR-10 [16] with more sophisticated noise transition matrices. Considering that both datasets have 10 classes, we adopt the following procedure to generate the $10 \times 10$ noise transition matrix: (1) manually set the diagonal elements at least $0.4$; (2) permute the diagonal elements to increase the randomness; (3) fill out the non-diagonal elements randomly and ensure the sum of each column is 1. An MLP model is trained from scratch on MNIST dataset, while a pre-trained vision transformer [8] is used to extract visual features on CIFAR-10 dataset. As shown in Table 2, we observe that when the noise gap is $0.1$ and $0.2$, CE and Peer with NOISE+ outperforms the pure CE and Peer, respectively. When the noise gap is $0.3$, balancing cannot compensate for the performance drop due to increased noise.

**Experiments on Clothing1M.** Clothing1M is a large-scale dataset collected from online shopping websites, comprising of 1 million training images of clothes with realistic noisy and 10,000 test data

Table 4: **Constrained learning results with group-dependent label noise.** We only report the results for $e_a = 0.2$ and $e_b = 0.4$. LR: naïve logistic regression without noise correction. GPR: group-weighted peer loss [30]. Peer: peer loss [21]. We highlight numbers of low fairness violation.

| Dataset | Metrics | LESS NOISE | | MORE NOISE | |
|---------|---------|-----|-----|-----|------|
| | | LR | GPL | LR | Peer |
| Adult | *accuracy* | 72.73 | 71.2 | 71.88 | 73.02 |
| | *fairness* | 6.48 | 2.95 | 3.16 | 1.67 |
| Compas | *accuracy* | 62.60 | 66.03 | 66.22 | 64.15 |
| | *fairness* | 2.87 | 7.55 | 6.07 | 3.63 |
| Fairface | *accuracy* | 86.97 | 87.47 | 88.19 | 87.93 |
| | *fairness* | 5.87 | 4.70 | 1.38 | 0.25 |

with clean labels. For a fair comparison, we adopted the same setting as described in [27] and trained a ResNet-50 [13] classifier. Table 3 compares our methods with some other baselines, including loss correction [27] and Co-Teaching [11]. Without a careful tuning of training parameters, vanilla CE with NOISE+reached 70.37% test accuracy. In comparison, standard CE achieves 68.94%, loss correction achieves 69.84%, and Co-Teaching is 70.15%.

## 6.2 Constrained learning

**Setup.** We conduct experiments with equal odds constraints [12] on Adult, Compas, and Fairface datasets. We add heterogeneous noise, i.e., uneven noise rates for different group memberships but symmetric for different classes, into training labels but keep test labels clean. We stress-test the performance of the naïve logistic regression (LR) and group-weighted peer loss (GPL) [30] with and without our increasing-to-balancing program.[1] We note that GPL degrades to peer loss when noise rates are homogeneous across protected groups [30], and directly apply peer loss after balancing. For a fair comparison, we use logistic regression as the base classifier for all the methods. We use the reductions approach [1] to solve the constrained optimization.

**Results.** We report the accuracy and fairness violation for $e_a = 0.2$ and $e_b = 0.4$ in Table 4 and defer more noise settings to Appendix **??**. We make the following observations: (1) All the methods retain a comparable accuracy. We conjecture that this benefits from the symmetric noise rates between positive and negative classes. (2) After balancing, both naïve LR and Peer significantly mitigate the fairness violations on Adult and Fairface datasets as a result of equalized noise rates across groups. (3) Peer loss generally has a lower fairness violation than naïve LR. This implies its capability of recovering unbiased classifiers. (4) We find out that the performance of LR with increasing-to-balancing is less profitable on Compas dataset because of uninformative features (only 10-dimensional features), and more robust on Fairface dataset with rich image information.

## 7 Concluding Remarks

This paper elaborates the possibility of improving the training model accuracy and fairness guarantees by increasing a certain class of instances' noise rates. Our idea is based on several observations that increasing-to-balance label noise rates can yield an easier learning problem that is robust to mis-specified noise rate parameters. The above robustness helps us improve generalization power as well as fairness guarantees. To deploy our idea, we propose a detection algorithm to identify the class of instances with a higher label noise rate, without using any ground truth label information.

Our noise rate balancing solution is primarily a data pre-processing procedure and is compatible with most existing learning with noisy label solutions. Our experiment result using peer loss is one such example. Empirically we do observe that many other solutions may also enjoy the benefits of balancing the noise rates. We believe this data pre-processing technique has the potential to find applications in other learning tasks when balancing label noise or bias is desired.

---

[1]We did not repeat testing surrogate loss as it is reported in [30] that GPL appears to be a better and more robust solution. Also we focus on correcting fairness constraints in this section, both SL and GPL would use the same constraint corrections.

## Acknowledgments

This work is partially supported by the National Science Foundation (NSF) under grant IIS-2007951 and the NSF FAI program in collaboration with Amazon under grant IIS-2040800. The authors would like to thank Jiaheng Wei and Zhaowei Zhu for their prompt discussion and help on the experimental setup for the Clothing1M dataset.

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
