# Supplementary Material
# Can Less be More? When Increasing-to-Balancing Label Noise Rates Considered Beneficial

**Yang Liu**
Computer Science and Engineering
University of California, Santa Cruz
Santa Cruz, CA 95064
yangliu@ucsc.edu

**Jialu Wang**
Computer Science and Engineering
University of California, Santa Cruz
Santa Cruz, CA 95064
faldict@ucsc.edu

This Appendix is organized as follows:

- Section A includes omitted proofs for theoretical conclusions in the main paper, as well as the extension to fairness constrained setting (A.9) and multi-class classification (A.10).
- Section B presents more experimental details and results.

## A    Omitted Proofs

### A.1    Proof for Theorem 1

**Proof** Let $\ell^\diamond$ denote the noise-corrected loss with respect to true noise parameters $e_+, e_-$:

$$\ell^\diamond(h(x_n), \tilde{y}_n) := (1 - e_{-sgn(\tilde{y}_n)}) \cdot \ell(h(x_n), \tilde{y}_n) - e_{sgn(\tilde{y}_n)} \cdot \ell(h(x_n), -\tilde{y}_n) \tag{A1}$$

It was established in [25] the unbiasedness of $\ell^\diamond$:

**Lemma 11** (Unbiasedness of $\ell^\diamond$, [25]). $\frac{1}{1-e_+-e_-} \cdot \mathbb{E}_{\tilde{Y}|Y=y}[\ell^\diamond(h(x), \tilde{Y})] = \ell(h(x), y)$.

A direct consequence of this lemma, via repeatedly applying to each $(X, Y)$, is its unbiasedness on the population level:

$$\frac{1}{1 - e_+ - e_-} \cdot \mathbb{E}_{\tilde{D}|D}[\hat{R}_{\ell^\diamond, \tilde{D}}(h)] = \hat{R}_{\ell, D}(h), \quad \frac{1}{1 - e_+ - e_-} \cdot R_{\ell^\diamond, \tilde{\mathcal{D}}}(h) = R_{\ell, \mathcal{D}}(h)$$

The following fact holds by subtracting $\ell^\diamond$ from $\tilde{\ell}$:

$$\tilde{\ell}(h(x_n), \tilde{y}_n) = \ell^\diamond(h(x_n), \tilde{y}_n) + (e_{-\tilde{y}_n} - \tilde{e}_{-\tilde{y}_n}) \cdot \ell(h(x_n), \tilde{y}_n) + (\tilde{e}_{-\tilde{y}_n} - e_{\tilde{y}_n}) \cdot \ell(h(x_n), -\tilde{y}_n)$$

Using the triangle inequality of $|\cdot|$ we establish that

$$|\tilde{\ell}(h(x_n), \tilde{y}_n) - \ell^\diamond(h(x_n), \tilde{y}_n)| \leq \max\{|\tilde{e}_+ - e_+|, |\tilde{e}_- - e_-|\} \cdot \bar{\ell}. \tag{A2}$$

This further helps us bound the differences in the empirical risks:

$$|\hat{R}_{\tilde{\ell}, \tilde{D}}(h) - \hat{R}_{\ell^\diamond, \tilde{D}}(h)| \leq \max\{|\tilde{e}_+ - e_+|, |\tilde{e}_- - e_-|\} \cdot \bar{\ell} \tag{A3}$$

Therefore

$$\hat{R}_{\ell^\diamond, \tilde{D}}(h^*_{\tilde{\ell}, \tilde{D}}) \leq \hat{R}_{\tilde{\ell}, \tilde{D}}(h^*_{\tilde{\ell}, \tilde{D}}) + \max\{|\tilde{e}_+ - e_+|, |\tilde{e}_- - e_-|\} \cdot \bar{\ell}$$

$$\leq \hat{R}_{\tilde{\ell}, \tilde{D}}(h^*_{\ell, \mathcal{D}}) + \max\{|\tilde{e}_+ - e_+|, |\tilde{e}_- - e_-|\} \cdot \bar{\ell} \qquad \text{(Opt. of } h^*_{\ell, \hat{D}})$$

$$\leq \hat{R}_{\ell^\diamond, \tilde{D}}(h^*_{\ell, \mathcal{D}}) + 2\max\{|\tilde{e}_+ - e_+|, |\tilde{e}_- - e_-|\} \cdot \bar{\ell} \tag{A4}$$

Calling the results in [25], [Rademacher bound for max risk deviation, Proof of Lemma 2 therein], we know that for any $\delta > 0$, with probability at least $1 - \delta$:

$$\sup_{h \in \mathcal{H}} \frac{1}{1 - e_+ - e_-} \left| R_{\ell^\diamond, \tilde{\mathcal{D}}}(h) - \hat{R}_{\ell^\diamond, \tilde{D}}(h) \right| \leq \frac{2L}{1 - e_+ - e_-} \cdot \mathcal{R}(\mathcal{H}) + \sqrt{\frac{\log 1/\delta}{2N}} \quad \text{(A5)}$$

The above knowledge further helps us establish that

$$R_{\ell, \mathcal{D}}(h^*_{\tilde{\ell}, \tilde{D}}) - R_{\ell, \mathcal{D}}(h^*_{\ell, \mathcal{D}})$$

$$= \frac{1}{1 - e_+ - e_-} (R_{\ell^\diamond, \tilde{\mathcal{D}}}(h^*_{\tilde{\ell}, \tilde{D}}) - R_{\ell^\diamond, \tilde{\mathcal{D}}}(h^*_{\ell, \mathcal{D}})) \qquad \text{(Unbiasedness of } \ell^\diamond \text{ on } \tilde{\mathcal{D}})$$

$$= \frac{1}{1 - e_+ - e_-} (R_{\ell^\diamond, \tilde{\mathcal{D}}}(h^*_{\tilde{\ell}, \tilde{D}}) - \hat{R}_{\ell^\diamond, \tilde{D}}(h^*_{\tilde{\ell}, \tilde{D}})) \qquad \text{(Rademacher bound )}$$

$$+ \frac{1}{1 - e_+ - e_-} (\hat{R}_{\ell^\diamond, \tilde{D}}(h^*_{\tilde{\ell}, \tilde{D}}) - \hat{R}_{\ell^\diamond, \tilde{D}}(h^*_{\ell, \mathcal{D}})) \qquad \text{(Eqn. (A4))}$$

$$+ \frac{1}{1 - e_+ - e_-} (\hat{R}_{\ell^\diamond, \tilde{D}}(h^*_{\ell, \mathcal{D}}) - R_{\ell^\diamond, \tilde{\mathcal{D}}}(h^*_{\ell, \mathcal{D}})) \qquad \text{(Rademacher bound )}$$

$$\leq \frac{4L}{1 - e_+ - e_-} \cdot \mathcal{R}(\mathcal{H}) + 2\sqrt{\frac{\log 1/\delta}{2N}} + 2\frac{\max\{|\tilde{e}_+ - e_+|, |\tilde{e}_- - e_-|\}}{1 - e_+ - e_-} \cdot \bar{\ell}$$

$$\leq \frac{4L}{1 - e_+ - e_-} \cdot \mathcal{R}(\mathcal{H}) + 2\sqrt{\frac{\log 1/\delta}{2N}} + 2\frac{\texttt{err}_{\texttt{M}}}{1 - e_+ - e_-} \cdot \bar{\ell} .$$

We complete the proof. ∎

## A.2 Proof for Lemma 2

**Proof** Expanding $\mathbb{P}(h(X) \neq \hat{Y})$ using the law of total probability we have

$$\mathbb{P}(h(X) \neq \hat{Y}) = \mathbb{P}(h(X) \neq \hat{Y}, \hat{Y} \neq Y) + \mathbb{P}(h(X) \neq \hat{Y}, \hat{Y} = Y)$$
$$= \mathbb{P}(h(X) \neq \hat{Y} \mid \hat{Y} \neq Y) \cdot \mathbb{P}(\hat{Y} \neq Y) + \mathbb{P}(h(X) \neq \hat{Y} \mid \hat{Y} = Y) \cdot \mathbb{P}(\hat{Y} = Y).$$

In binary classification, $h(X) \neq \tilde{Y}, \tilde{Y} \neq Y$ implies $h(X) = Y$, such that

$$\mathbb{P}(h(X) \neq \tilde{Y} \mid \tilde{Y} \neq Y) = \mathbb{P}(h(X) = Y \mid \tilde{Y} \neq Y).$$

Due to the independence of $\tilde{Y}$ and $X$ given $Y$,

$$\mathbb{P}(h(X) = Y \mid \tilde{Y} \neq Y) = \frac{\mathbb{P}(h(X) = Y, \tilde{Y} \neq Y)}{\mathbb{P}(\tilde{Y} \neq Y)} = \frac{\mathbb{P}(h(X) = Y)\mathbb{P}(\tilde{Y} \neq Y)}{\mathbb{P}(\tilde{Y} \neq Y)} = \mathbb{P}(h(X) = Y)$$

Similarly, we have

$$\mathbb{P}(h(X) \neq \hat{Y} \mid \hat{Y} = Y) = \mathbb{P}(h(X) \neq \tilde{Y}).$$

Combining all above we finished the proof when $e < 0.5$ by having:

$$\mathbb{P}(h(X) \neq \hat{Y}) = \mathbb{P}(h(X) = Y) \cdot e + \mathbb{P}(h(X) \neq Y) \cdot (1 - e)$$
$$= (1 - 2e) \cdot \mathbb{P}(h(X) \neq Y) + e$$

∎

## A.3 Proof for Theorem 3

**Proof** Again let $\ell^\diamond$ denote the noise-corrected loss with respect to true noise parameters $e_+, e_-$:

$$\ell^\diamond(h(x_n), \tilde{y}_n) := (1 - e_{-sgn(\tilde{y}_n)}) \cdot \ell(h(x_n), \tilde{y}_n) - e_{sgn(\tilde{y}_n)} \cdot \ell(h(x_n), -\tilde{y}_n) \quad \text{(A6)}$$

First notice the following when $\ell$ is a symmetric loss:

$$R_{\ell, \mathcal{D}}(h^*_{\ell, \hat{D}})$$

$$= \frac{1}{1 - 2e} \cdot R_{\ell^\diamond, \hat{\mathcal{D}}}(h^*_{\ell, \hat{D}}) \qquad \text{(Unbiasedness of } \ell^\diamond \text{ on } \hat{\mathcal{D}} \text{ using symmetric } e)$$

$$= \frac{1 - e}{1 - 2e} \cdot R_{\ell, \hat{\mathcal{D}}}(h^*_{\ell, \hat{D}}) - \frac{e}{1 - 2e} \cdot R_{\ell, \hat{\mathcal{D}}}(-h^*_{\ell, \hat{D}}) \qquad \text{(A7)}$$

The last equality uses the definition of $\ell^\diamond$, and is due to $\ell$ being symmetric. Then we show that

$$\frac{1}{1-2e} \cdot \left( R_{\ell,\hat{\mathcal{D}}}(h^*_{\ell,\hat{D}}) - R_{\ell,\hat{\mathcal{D}}}(h^*_{\ell,\mathcal{D}}) \right)$$

$$= \frac{1}{1-2e} \left( R_{\ell,\hat{\mathcal{D}}}(h^*_{\ell,\hat{D}}) - \hat{R}_{\ell,\hat{D}}(h^*_{\ell,\hat{D}}) \right) \qquad \text{(Rademacher bound )}$$

$$+ \frac{1}{1-2e} \left( \hat{R}_{\ell,\hat{D}}(h^*_{\ell,\hat{D}}) - \hat{R}_{\ell,\hat{D}}(h^*_{\ell,\mathcal{D}}) \right) \qquad (\leq 0 \ \text{ Optimality of } h^*_{\ell,\hat{D}} \text{ on } \ell, \hat{D})$$

$$+ \frac{1}{1-2e} \left( \hat{R}_{\ell,\hat{D}}(h^*_{\ell,\mathcal{D}}) - R_{\ell,\hat{\mathcal{D}}}(h^*_{\ell,\mathcal{D}}) \right) \qquad \text{(Rademacher bound )}$$

$$\leq \frac{4L}{1-2e}\mathcal{R}(\mathcal{H}) + 2\sqrt{\frac{\log 1/\delta}{2N}}$$

The inequality is due to the Rademacher bound we invoked as in Eqn. (A5) as well as the optimality of $h^*_{\ell,\hat{D}}$ on $\ell, \hat{D}$. That is we have proved with probability at least $1 - \delta$ that

$$\frac{1}{1-2e} \cdot R_{\ell,\hat{\mathcal{D}}}(h^*_{\ell,\hat{D}}) \leq \frac{1}{1-2e} \cdot R_{\ell,\hat{\mathcal{D}}}(h^*_{\ell,\mathcal{D}}) + \frac{4L}{1-2e}\mathcal{R}(\mathcal{H}) + 2\sqrt{\frac{\log 1/\delta}{2N}} \qquad \text{(A8)}$$

Repeating the same analysis and using the assumed condition that $\hat{R}_{\ell,\hat{D}}(-h^*_{\ell,\hat{D}}) - \hat{R}_{\ell,\hat{D}}(-h^*_{\ell,\mathcal{D}}) \geq 0$ we have

$$\frac{1}{1-2e} \cdot R_{\ell,\hat{\mathcal{D}}}(-h^*_{\ell,\hat{D}}) \geq \frac{1}{1-2e} \cdot R_{\ell,\hat{\mathcal{D}}}(-h^*_{\ell,\mathcal{D}}) - \frac{4L}{1-2e}\mathcal{R}(\mathcal{H}) - 2\sqrt{\frac{\log 1/\delta}{2N}} \qquad \text{(A9)}$$

Combining above (Eqn. (A8) and (A9)), we have with probability at least $1 - \delta$ (that both of the above bounds will happen simultaneously)

$$R_{\ell,\mathcal{D}}(h^*_{\ell,\hat{D}}) = \frac{1-e}{1-2e} \cdot R_{\ell,\hat{\mathcal{D}}}(h^*_{\ell,\hat{D}}) - \frac{e}{1-2e} \cdot R_{\ell,\hat{\mathcal{D}}}(-h^*_{\ell,\hat{D}})$$

$$\leq \frac{1-e}{1-2e} \cdot R_{\ell,\hat{\mathcal{D}}}(h^*_{\ell,\mathcal{D}}) - \frac{e}{1-2e} \cdot R_{\ell,\hat{\mathcal{D}}}(-h^*_{\ell,\mathcal{D}}) + \frac{4L}{1-2e}\mathcal{R}(\mathcal{H}) + 2\sqrt{\frac{\log 1/\delta}{2N}}$$

$$= R_{\ell,\mathcal{D}}(h^*_{\ell,\mathcal{D}}) + \frac{4L}{1-2e}\mathcal{R}(\mathcal{H}) + 2\sqrt{\frac{\log 1/\delta}{2N}}.$$

The inequality uses Eqn. (A8) and (A9). Again the last equality is reusing Eqn. (A7). This completes the proof. ∎

### A.4  Proof for Lemma 4

**Proof**

$$\mathbb{P}(h(X) = +1|\tilde{Y} = +1, Z = a) = \frac{\mathbb{P}(h(X) = +1, \tilde{Y} = +1|Z = a)}{\mathbb{P}(\tilde{Y} = +1|Z = a)} \qquad \text{(A10)}$$

Again we do the trick of sampling $\mathbb{P}(\tilde{Y} = +1|Z = a)$ to be 0.5, which allows us to focus on the numerator.

$$\mathbb{P}(h(X) = +1, \tilde{Y} = +1|Z = a)$$

$$= \mathbb{P}(h(X) = +1, \tilde{Y} = +1, Y = +1|Z = a)$$

$$+ \mathbb{P}(h(X) = +1, \tilde{Y} = +1, Y = -1|Z = a)$$

$$= \mathbb{P}(h(X) = +1, \tilde{Y} = +1|Y = +1, Z = a) \cdot \mathbb{P}(Y = +1|Z = a)$$

$$+ \mathbb{P}(h(X) = +1, \tilde{Y} = +1|Y = -1, Z = a) \cdot \mathbb{P}(Y = -1|Z = a)$$

$$= \mathbb{P}(h(X) = +1|Y = +1, Z = a) \cdot (1 - e_a) \cdot \mathbb{P}(Y = +1|Z = a)$$

$$+ \mathbb{P}(h(X) = +1|Y = -1, Z = a) \cdot e_a \cdot \mathbb{P}(Y = -1|Z = a)$$

$$\text{(Independence of } X \text{ and } \tilde{Y} \text{ given } Y)$$

That is

$$0.5 \cdot \widetilde{\mathrm{TPR}}_a(h) = \mathrm{TPR}_a(h) \cdot (1 - e_a) \cdot \mathbb{P}(Y = +1 | Z = a) + \mathrm{FPR}_a(h) \cdot e_a \cdot \mathbb{P}(Y = -1 | Z = a)$$
(A11)

Similarly for FPR we have

$$\mathbb{P}(h(X) = +1 | \tilde{Y} = -1, Z = a) = \frac{\mathbb{P}(h(X) = +1, \tilde{Y} = -1 | Z = a)}{\mathbb{P}(\tilde{Y} = -1 | Z = a)}$$
(A12)

Following similar steps as above, the numerator further derives as

$$\begin{aligned}
& \mathbb{P}(h(X) = +1, \tilde{Y} = +1 | Z = a) \\
={}& \mathbb{P}(h(X) = +1 | Y = -1, Z = a) \cdot (1 - e_a) \cdot \mathbb{P}(Y = +1 | Z = a) \\
& + \mathbb{P}(h(X) = +1 | Y = +1, Z = a) \cdot e_a \cdot \mathbb{P}(Y = -1 | Z = a)
\end{aligned}$$

That is

$$0.5 \cdot \widetilde{\mathrm{FPR}}_a(h) = \mathrm{FPR}_a(h) \cdot (1 - e_a) \cdot \mathbb{P}(Y = +1 | Z = a) + \mathrm{TPR}_a(h) \cdot e_a \cdot \mathbb{P}(Y = -1 | Z = a)$$
(A13)

When $\mathbb{P}(\tilde{Y} = +1 | Z = a) = \mathbb{P}(\tilde{Y} = +1 | Z = b) = 0.5$, we will also have

$$0.5 = \mathbb{P}(\tilde{Y} = +1 | Z = a) = \mathbb{P}(Y = +1 | Z = a)(1 - e_a) + \mathbb{P}(Y = -1 | Z = a)e_a$$
(A14)

which returns us that $\mathbb{P}(Y = +1 | Z = a) = \frac{0.5 - e_a}{1 - 2e_a} := p = 0.5$. Using this knowledge and solving the linear equations defined by Eqn. (A11) and (A13) we have

$$\mathrm{TPR}_a(h) = \frac{C_{a,1} \cdot \widetilde{\mathrm{TPR}}_a(h) - C_{a,2} \cdot \widetilde{\mathrm{FPR}}_a(h)}{e_a - 0.5}$$
(A15)

$$\mathrm{FPR}_a(h) = \frac{C_{a,1} \cdot \widetilde{\mathrm{FPR}}_a(h) - C_{a,2} \cdot \widetilde{\mathrm{TPR}}_a(h)}{e_a - 0.5}$$
(A16)

∎

## A.5 Proof for Theorem 5

**Proof** Combining Eqn. (5) and (6) we have

$$\begin{aligned}
& |\mathrm{TPR}_z(h) - \mathrm{TPR}_z^c(h)| \\
={}& \left| \frac{0.5 \cdot e_z \cdot \widetilde{\mathrm{TPR}}_z(h) - 0.5(1 - e_z) \cdot \widetilde{\mathrm{FPR}}_z(h)}{e_z - 0.5} - \frac{0.5 \cdot \tilde{e}_z \cdot \widetilde{\mathrm{TPR}}_z(h) - 0.5(1 - \tilde{e}_z) \cdot \widetilde{\mathrm{FPR}}_z(h)}{\tilde{e}_z - 0.5} \right| \\
={}& \frac{|\tilde{e}_z - e_z| \cdot \widetilde{\mathrm{TPR}}_z(h)}{(2e_z - 1)(2\tilde{e}_z - 1)} \\
={}& \frac{\mathrm{err}_z \cdot \widetilde{\mathrm{TPR}}_z(h)}{(2e_z - 1)(2\tilde{e}_z - 1)} \; .
\end{aligned}$$
(A17)

Recall $\mathrm{err}_z = |\tilde{e}_z - e_z|$. The second equality is algebraic - we simply unify the denominator of both quantities and rearrange terms. Then equalizing TPR that $\mathrm{TPR}_a^c(h) = \mathrm{TPR}_b^c(h)$ returns us

$$\begin{aligned}
& |\mathrm{TPR}_a(h) - \mathrm{TPR}_b(h)| \\
={}& |\mathrm{TPR}_a(h) - \mathrm{TPR}_a^c(h) + \mathrm{TPR}_b^c(h) - \mathrm{TPR}_b(h)| \\
\geq{}& ||\mathrm{TPR}_a(h) - \mathrm{TPR}_a^c(h)| - |\mathrm{TPR}_b^c(h) - \mathrm{TPR}_b(h)|| \\
={}& \left| \frac{\mathrm{err}_a \cdot \widetilde{\mathrm{TPR}}_a(h)}{(2e_a - 1)(2\tilde{e}_a - 1)} - \frac{\mathrm{err}_b \cdot \widetilde{\mathrm{TPR}}_b(h)}{(2e_b - 1)(2\tilde{e}_b - 1)} \right| ,
\end{aligned}$$

where the last equality is an application of Eqn. (A17). Then

$$\left| \frac{\mathrm{err}_a \cdot \widetilde{\mathrm{TPR}}_a(h)}{(2e_a-1)(2\tilde{e}_a-1)} - \frac{\mathrm{err}_b \cdot \widetilde{\mathrm{TPR}}_b(h)}{(2e_b-1)(2\tilde{e}_b-1)} \right|$$

$$= \mathrm{err}_a \cdot \left| \frac{\widetilde{\mathrm{TPR}}_a(h)}{(2e_a-1)(2\tilde{e}_a-1)} - \frac{\mathrm{err}_b}{\mathrm{err}_a} \frac{\widetilde{\mathrm{TPR}}_b(h)}{(2e_b-1)(2\tilde{e}_b-1)} \right|$$

$$\geq \mathrm{err}_\mathrm{M} \cdot \left| \frac{\widetilde{\mathrm{TPR}}_a(h)}{(2e_a-1)(2\tilde{e}_a-1)} - \frac{\mathrm{err}_b}{\mathrm{err}_a} \frac{\widetilde{\mathrm{TPR}}_b(h)}{(2e_b-1)(2\tilde{e}_b-1)} \right|$$

Similarly

$$|\mathrm{FPR}_z(h) - \mathrm{FPR}_z^c(h)|$$

$$= \left| \frac{0.5 \cdot e_z \cdot \widetilde{\mathrm{FPR}}_z(h) - 0.5(1-e_z) \cdot \widetilde{\mathrm{TPR}}_z(h)}{e_z - 0.5} - \frac{0.5 \cdot \tilde{e}_z \cdot \widetilde{\mathrm{FPR}}_z(h) - 0.5(1-\tilde{e}_z) \cdot \widetilde{\mathrm{TPR}}_z(h)}{\tilde{e}_z - 0.5} \right|$$

$$= \frac{|\tilde{e}_z - e_z| \cdot \widetilde{\mathrm{FPR}}_z(h)}{(2e_z-1)(2\tilde{e}_z-1)}.$$

$$= \frac{\mathrm{err}_z \cdot \widetilde{\mathrm{FPR}}_z(h)}{(2e_z-1)(2\tilde{e}_z-1)}.$$

Then equalizing FPR that $\mathrm{FPR}_a^c(h) = \mathrm{FPR}_b^c(h)$ we have

$$|\mathrm{FPR}_a(h) - \mathrm{FPR}_b(h)|$$
$$= |\mathrm{FPR}_a(h) - \mathrm{FPR}_a^c(h) + \mathrm{FPR}_b^c(h) - \mathrm{FPR}_b(h)|$$
$$\geq ||\mathrm{FPR}_a(h) - \mathrm{FPR}_a^c(h)| - |\mathrm{FPR}_b^c(h) - \mathrm{FPR}_b(h)||$$
$$\geq \left| \frac{\mathrm{err}_a \cdot \widetilde{\mathrm{FPR}}_a(h)}{(2e_a-1)(2\tilde{e}_a-1)} - \frac{\mathrm{err}_b \cdot \widetilde{\mathrm{FPR}}_b(h)}{(2e_b-1)(2\tilde{e}_b-1)} \right|$$
$$\geq \mathrm{err}_\mathrm{M} \cdot \left| \frac{\widetilde{\mathrm{FPR}}_a(h)}{(2e_a-1)(2\tilde{e}_a-1)} - \frac{\mathrm{err}_b}{\mathrm{err}_a} \frac{\widetilde{\mathrm{FPR}}_b(h)}{(2e_b-1)(2\tilde{e}_b-1)} \right|.$$

∎

## A.6 Proof for Theorem 6

**Proof** Easy to show that when $e_a = e_b$, $C_{a,1} = C_{b,1}$ and $C_{a,2} = C_{b,2}$. Therefore, from Eqn. (5) we know equalizing

$$\widetilde{\mathrm{TPR}}_a(h) = \widetilde{\mathrm{TPR}}_b(h), \quad \widetilde{\mathrm{FPR}}_a(h) = \widetilde{\mathrm{FPR}}_b(h) \tag{A18}$$

will also return us

$$\mathrm{TPR}_a(h) = \mathrm{TPR}_b(h), \quad \mathrm{FPR}_a(h) = \mathrm{FPR}_b(h) \tag{A19}$$

∎

## A.7 Proof for Theorem 9

**Proof** We start with deriving $\mathrm{PA}_{\mathcal{D}^\diamond}$.

$$\mathrm{PA}_{\mathcal{D}^\diamond} = \mathbb{P}(\tilde{Y}_2 = \tilde{Y}_3 = +1 | \tilde{Y}_1 = +1) = \frac{\mathbb{P}(\tilde{Y}_1 = \tilde{Y}_2 = \tilde{Y}_3 = +1)}{\mathbb{P}(\tilde{Y}_1 = +1)}$$

Due to the sampling step, we have $\mathbb{P}(\tilde{Y}_1 = +1) = 0.5$ - this allows us to focus on the denominator:

$$\mathbb{P}(\tilde{Y}_1 = \tilde{Y}_2 = \tilde{Y}_3 = +1) \overset{(1)}{=} \mathbb{P}(Y=+1) \prod_{i=1}^{3} \mathbb{P}(\tilde{Y}_i = +1 | Y=+1) + \mathbb{P}(Y=-1) \prod_{i=1}^{3} \mathbb{P}(\tilde{Y}_i = +1 | Y=-1)$$

$$\overset{(2)}{=} \mathbb{P}(Y=+1) \cdot (1-e_+)^3 + \mathbb{P}(Y=-1) \cdot e_-^3$$

where in above, (1) uses the 2-NN clusterability of $D$, and (2) uses the conditional independence between the noisy labels. Similarly for $\text{NA}_{\mathcal{D}^\diamond}$ we have:

$$\mathbb{P}(\tilde{Y}_2 = \tilde{Y}_3 = -1|\tilde{Y}_1 = -1) = \frac{\mathbb{P}(\tilde{Y}_1 = \tilde{Y}_2 = \tilde{Y}_3 = -1)}{\mathbb{P}(\tilde{Y}_1 = -1)}$$

Again we have that $\mathbb{P}(\tilde{Y}_1 = -1) = 0.5$, and the numerator derives as

$$\mathbb{P}(\tilde{Y}_1 = \tilde{Y}_2 = \tilde{Y}_3 = -1) = \mathbb{P}(Y = +1)\prod_{i=1}^{3}\mathbb{P}(\tilde{Y}_i = -1|Y = +1) + \mathbb{P}(Y = -1)\prod_{i=1}^{3}\mathbb{P}(\tilde{Y}_i = -1|Y = -1)$$

$$= \mathbb{P}(Y = +1)\cdot e_+^3 + \mathbb{P}(Y = -1)\cdot(1 - e_-)^3$$

Taking the difference (and normalize by 0.5) we have

$$0.5\cdot(\text{PA}_{\mathcal{D}^\diamond} - \text{NA}_{\mathcal{D}^\diamond})$$
$$= \mathbb{P}(\tilde{Y}_2 = \tilde{Y}_3 = +1|\tilde{Y}_1 = +1) - \mathbb{P}(\tilde{Y}_2 = \tilde{Y}_3 = -1|\tilde{Y}_1 = -1)$$
$$= \mathbb{P}(Y = +1)\left((1 - e_+)^3 - e_+^3\right) + \mathbb{P}(Y = -1)\left(e_-^3 - (1 - e_-)^3\right) \qquad (A20)$$

Notice two facts: first we can derive that

$$(1 - e_+)^3 - e_+^3 = (1 - 2e_+)(e_+^2 - e_+ + 1), \;\; e_-^3 - (1 - e_-)^3 = -(1 - 2e_-)(e_-^2 - e_- + 1)$$

Second, we will use the following fact:

$$0.5 = \mathbb{P}(\tilde{Y} = +1) = \mathbb{P}(Y = +1)(1 - e_+) + \mathbb{P}(Y = -1)e_- \qquad (A21)$$

from which we solve that $\mathbb{P}(Y = +1) = \frac{0.5 - e_-}{1 - e_+ - e_-}$. Symmetrically, $\mathbb{P}(Y = -1) = \frac{0.5 - e_+}{1 - e_+ - e_-}$.

Return the above two facts back into Eqn. (A20), we have

$$\mathbb{P}(Y = +1)((1 - e_+)^3 - e_+^3) + \mathbb{P}(Y = -1)(e_-^3 - (1 - e_-)^3)$$
$$= 2\cdot\frac{(0.5 - e_+)(0.5 - e_-)}{1 - e_+ - e_-}\left((e_+^2 - e_+ + 1) - (e_-^2 - e_- + 1)\right)$$
$$= 2\cdot(0.5 - e_+)\cdot(0.5 - e_-)\cdot(e_- - e_+)$$

completing the proof when $e_+, e_- < 0.5$. ∎

## A.8 Proof for Proposition 10

**Proof** Expanding $\mathbb{P}(\hat{Y} = -1|Y = +1)$ using the law of total probability we have

$$\hat{e}_+ = \mathbb{P}(\hat{Y} = -1|Y = +1)$$
$$= \mathbb{P}(\hat{Y} = -1, \tilde{Y} = +1|Y = +1) + \mathbb{P}(\hat{Y} = -1, \tilde{Y} = -1|Y = +1)$$
$$= \mathbb{P}(\hat{Y} = -1|\tilde{Y} = +1, Y = +1)\cdot\mathbb{P}(\tilde{Y} = +1|Y = +1)$$
$$\quad + \mathbb{P}(\hat{Y} = -1|\tilde{Y} = -1, Y = +1)\cdot\mathbb{P}(\tilde{Y} = -1|Y = +1)$$
$$= \epsilon\cdot(1 - e_+) + 1\cdot e_+ \qquad \text{(Independence between } \hat{Y} \text{ and } Y \text{ given } \tilde{Y})$$
$$= (1 - e_+)\cdot\epsilon + e_+$$

Similarly,

$$\hat{e}_- = \mathbb{P}(\hat{Y} = +1|Y = -1)$$
$$= \mathbb{P}(\hat{Y} = +1, \tilde{Y} = +1|Y = -1) + \mathbb{P}(\hat{Y} = +1, \tilde{Y} = -1|Y = -1)$$
$$= \mathbb{P}(\hat{Y} = +1|\tilde{Y} = +1, Y = -1)\cdot\mathbb{P}(\tilde{Y} = +1|Y = -1)$$
$$\quad + \mathbb{P}(\hat{Y} = +1|\tilde{Y} = -1, Y = -1)\cdot\mathbb{P}(\tilde{Y} = -1|Y = -1)$$
$$= (1 - \epsilon)\cdot e_-.$$

The last equality is again due to the independence between $\hat{Y}$ and $Y$ given $\tilde{Y}$, as well as the fact that we do not flip the $\tilde{Y} = -1$ labels so $\mathbb{P}(\hat{Y} = +1|\tilde{Y} = -1, Y = -1) = 0$. Taking the difference we finish the proof. ∎

## A.9 Balancing noise for fairness constrained case

Define

$$\text{PA}_{\mathcal{D}^\diamond,a} = \mathbb{P}_{Z=a}(\tilde{Y}_2 = \tilde{Y}_3 = +1|\tilde{Y}_1 = +1) \tag{A22}$$

$$\text{PA}_{\mathcal{D}^\diamond,b} = \mathbb{P}_{Z=b}(\tilde{Y}_2 = \tilde{Y}_3 = +1|\tilde{Y}_1 = +1) \tag{A23}$$

We now claim that $sgn(\text{PA}_{\mathcal{D}^\diamond,a} - \text{PA}_{\mathcal{D}^\diamond,b}) = -sgn(e_a - e_b)$. We start with deriving $\text{PA}_{\mathcal{D}^\diamond,a}$.

$$\text{PA}_{\mathcal{D}^\diamond,a} = \mathbb{P}_{Z=a}(\tilde{Y}_2 = \tilde{Y}_3 = +1|\tilde{Y}_1 = +1) = \frac{\mathbb{P}_{Z=a}(\tilde{Y}_1 = \tilde{Y}_2 = \tilde{Y}_3 = +1)}{\mathbb{P}_{Z=a}(\tilde{Y}_1 = +1)}$$

Due to the sampling step, we have $\mathbb{P}_{Z=a}(\tilde{Y}_1 = +1) = 0.5$ - this allows us to focus on the denominator:

$$\mathbb{P}_{Z=a}(\tilde{Y}_1 = \tilde{Y}_2 = \tilde{Y}_3 = +1) \overset{(1)}{=} \mathbb{P}_{Z=a}(Y = +1)\prod_{i=1}^{3}\mathbb{P}_{Z=a}(\tilde{Y}_i = +1|Y = +1)$$

$$+ \mathbb{P}_{Z=a}(Y = -1)\prod_{i=1}^{3}\mathbb{P}_{Z=a}(\tilde{Y}_i = +1|Y = -1)$$

$$\overset{(2)}{=} \mathbb{P}_{Z=a}(Y = +1) \cdot (1 - e_a)^3 + \mathbb{P}_{Z=a}(Y = -1) \cdot e_a^3$$

where in above, (1) uses the 2-NN clusterability of $D$, and (2) uses the conditional independence between the noisy labels. Similarly for $\text{PA}_{\mathcal{D}^\diamond,b}$ we have:

$$\text{PA}_{\mathcal{D}^\diamond,b} = \frac{\mathbb{P}_{Z=b}(Y = +1) \cdot (1 - e_b)^3 + \mathbb{P}_{Z=b}(Y = -1) \cdot e_b^3}{0.5} \tag{A24}$$

Firstly, we will use the following fact for $z \in \{a, b\}$:

$$0.5 = \mathbb{P}_{Z=z}(\tilde{Y} = +1)$$
$$= \mathbb{P}_{Z=z}(\tilde{Y} = +1|Y = +1) \cdot \mathbb{P}_{Z=z}(Y = +1) + \mathbb{P}_{Z=z}(\tilde{Y} = +1|Y = -1) \cdot \mathbb{P}_{Z=z}(Y = -1)$$
$$= \mathbb{P}_{Z=z}(Y = +1) \cdot (1 - e_z) + \mathbb{P}_{Z=z}(Y = -1) \cdot e_z$$

from which we solve that $\mathbb{P}_{Z=z}(Y = +1) = \frac{0.5 - e_z}{1 - 2e_z} = 0.5$. Therefore

$$\text{PA}_{\mathcal{D}^\diamond,a} - \text{PA}_{\mathcal{D}^\diamond,b} = (1 - e_a)^3 - (1 - e_b)^3 + e_a^3 - e_b^3$$
$$= (e_b - e_a)\left((1 - e_a)^2 + (1 - e_b)^2 + (1 - e_a)(1 - e_b) - e_a^2 - e_b^2 - e_a e_b\right)$$
$$= (e_b - e_a)(1 - 2e_a + 1 - 2e_b + 1 - e_a - e_b) \tag{A25}$$

Note that $1 - 2e_a + 1 - 2e_b + 1 - e_a - e_b > 0$ when $e_a, e_b < 0.5$. This implies that we can use the 2-NN positive agreements $\text{PA}_{\mathcal{D}^\diamond,a} - \text{PA}_{\mathcal{D}^\diamond,b}$ across groups to compare $e_a$ with $e_b$.

## A.10 Extension to multi-class

As explained at the beginning, our algorithm can largely extend to the multi-class/group setting. The primary requirement of the extension is to extend the definition of $\text{PA}_{\mathcal{D}^\diamond}, \text{NA}_{\mathcal{D}^\diamond}$ to each label class/group. Consider a multi-class classification problem with $K$ label classes, and the noise rates follow a uniform diagonal model:

$$\mathbb{P}(\tilde{Y} = k|Y = k) = 1 - e_k, \quad \mathbb{P}(\tilde{Y} = k'|Y = k) = \frac{e_k}{K - 1}, \quad \forall k' \neq k. \tag{A26}$$

Define $\text{KA}_{\mathcal{D}^\diamond,k} := \mathbb{P}(\tilde{Y}_2 = \tilde{Y}_3 = k|\tilde{Y}_1 = k), \ k = 1, 2, ..., K$. Similarly we can show that for any pair of $k_1, k_2$: $sgn(\text{KA}_{\mathcal{D}^\diamond,k_1} - \text{KA}_{\mathcal{D}^\diamond,k_2}) = -sgn(e_{k_1} - e_{k_2})$, wherein above $e_{k_1}, e_{k_2}$ are the error rates of label class $k_1, k_2$. With the above, we can compute $\text{KA}_{\mathcal{D}^\diamond,k}$, rank them, and start inserting noise to the classes that are determined to have a lower error rate to match the highest one.

## A.11 Pseudocodes

```
import numpy as np
from sklearn.neighbors import NearestNeighbors
def estimate_PA(X, y):
    nbrs = NearestNei.(n_neighbors=3, algorithm='ball_tree').fit(X)
    _, indices = nbrs.kneighbors(X)
    return np.mean(np.array([np.all(y[i] == y[indices[i]]) for i in np
        .where(y > 0)[0]]))
```

Figure A1: **Numpy-like pseudocode for an implementation of estimating PA.** Our implementation utilizes scikit-learn's Nearest Neighbors module. The code for esimating NA is similar.

## B  Additional Experiment Details and Results

We provide more details on the experimental setup as well as further results.

### B.1  Datasets

We evaluate our methods on five datasets:

- Adult, the UCI Adult Income dataset [9]. The task is to predict whether an individual's income exceeds 50K. The dataset consists of 48,842 examples and 28 features. We select female and male as two protected groups in constrained learning. We resample the dataset to ensure that both the classes and groups are balanced.

- Compas, the COMPAS recidivism dataset for crime statistics with 7,168 instances and 10 features [2]. We select race as the protected attribute in constrained learning.

- Fairface, the face attribute dataset containing 108,501 images with balanced race and gender groups [15]. We use a pre-trained vision transformer (ViT/B-32) model [8] to extract image representations, and project them into 50-dimensional feature vectors. For both unconstrained and constrained learning, we take gender attribute as labels for binary classification. For constrained learning, we categorize race into White and Non-White groups.

- MNIST [18], consisting of 50,000 training images and 10,000 test images in 10 classes. We train a MLP model from scratch on the MNIST dataset.

- CIFAR-10 [16], consisting of 50,000 training images and 10,000 test images in 10 classes. We evaluate unconstrained multi-class classification on CIFAR-10 dataset. Similar to Fairface, we use a pre-trained vision transformer to extract 512-dimensional feature vectors.

For Adult, Compas, and German datasets, we perform random train/test splits in a ratio of 80 to 20. For Fairface, MNIST, and CIFAR-10, we follow their original splits.

### B.2  Computing infrastructure

For all the experiments, we use a GPU cluster with 4 2080 Ti GPUs for training and evaluation.

### B.3  Noise transition matrix for CIFAR-10

We adopt the following procedure to generate the noise transition matrix:

1. Manually set the diagonal elements at least $0.4$. We ensure that the difference between the maximal elements and $0.4$ is equal to the noise gap.

2. Permute the diagonal elements to increase the randomness.

3. Fill out the non-diagonal elements randomly and ensure the sum of each column is 1

We show one sample noise transition matrix generated by our procedure with noise gap 0.2 as follows:

$$\begin{bmatrix}
\mathbf{0.4} & 0.087 & 0.013 & 0.032 & 0.032 & 0.068 & 0.050 & 0.178 & 0.001 & 0.118 \\
0.043 & \mathbf{0.4} & 0.002 & 0.016 & 0.049 & 0.113 & 0.060 & 0.024 & 0.224 & 0.017 \\
0.181 & 0.111 & \mathbf{0.4} & 0.147 & 0.033 & 0.005 & 0.026 & 0.040 & 0.110 & 0.076 \\
0.051 & 0.001 & 0.060 & \mathbf{0.6} & 0.032 & 0.047 & 0.149 & 0.145 & 0.022 & 0.059 \\
0.001 & 0.167 & 0.119 & 0.032 & \mathbf{0.6} & 0.092 & 0.051 & 0.018 & 0.037 & 0.129 \\
0.097 & 0.007 & 0.001 & 0.059 & 0.016 & \mathbf{0.4} & 0.019 & 0.014 & 0.084 & 0.001 \\
0.018 & 0.023 & 0.277 & 0.041 & 0.034 & 0.014 & \mathbf{0.4} & 0.028 & 0.041 & 0.062 \\
0.149 & 0.096 & 0.081 & 0.019 & 0.041 & 0.015 & 0.143 & \mathbf{0.4} & 0.061 & 0.110 \\
0.031 & 0.066 & 0.022 & 0.007 & 0.133 & 0.080 & 0.049 & 0.113 & \mathbf{0.4} & 0.025 \\
0.029 & 0.040 & 0.023 & 0.043 & 0.027 & 0.162 & 0.048 & 0.036 & 0.018 & \mathbf{0.4}
\end{bmatrix}$$

## B.4 Additional results

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

Table B2: **Accuracy of compared methods across different levels of noise gap for multi-class classification.** Mis. SL: surrogate loss [25] with misspecified parameters. Est. SL: surrogate loss [25] with estimated parameters. CE: vanilla cross entropy. Peer: peer loss function [21]. When noise gap is less than 0.2, cross entropy with increasing-to-balancing reaches a higher accuracy than cross entropy at a lower noise. When noise gap is 0.3, balancing cannot compensate for the loss of increasing noise.

| Dataset | noise gap | BASELINES (LESS NOISE) | | | | NOISE+ (MORE NOISE) | |
| | | Mis. SL | Est. SL | CE | Peer | CE | Peer |
|---|---|---|---|---|---|---|---|
| MNIST | 0.1 | $89.59 \pm 0.01$ | $89.69 \pm 0.07$ | $86.66 \pm 0.54$ | $88.12 \pm 0.01$ | $86.81 \pm 0.62$ | $89.19 \pm 0.05$ |
| | 0.2 | $88.10 \pm 0.10$ | $88.61 \pm 0.16$ | $84.53 \pm 1.60$ | $87.21 \pm 0.53$ | $85.97 \pm 0.69$ | $89.12 \pm 0.24$ |
| | 0.3 | $84.97 \pm 0.11$ | $86.88 \pm 0.17$ | $85.24 \pm 1.05$ | $86.35 \pm 0.33$ | $81.89 \pm 1.54$ | $88.75 \pm 0.19$ |
| CIFAR-10 | 0.1 | $70.90 \pm 2.66$ | $85.76 \pm 1.44$ | $88.03 \pm 1.07$ | $89.66 \pm 1.18$ | $88.69 \pm 0.82$ | $89.90 \pm 0.52$ |
| | 0.2 | $80.51 \pm 4.51$ | $86.34 \pm 2.30$ | $88.43 \pm 1.29$ | $89.36 \pm 0.56$ | $89.01 \pm 1.27$ | $90.08 \pm 1.26$ |
| | 0.3 | $81.30 \pm 2.31$ | $90.61 \pm 0.52$ | $89.78 \pm 1.16$ | $90.24 \pm 1.05$ | $87.98 \pm 1.29$ | $89.92 \pm 0.92$ |

Table B3: **Constrained learning results with group-dependent label noise.** LR: naïve logistic regression without noise correction. GPR: group-weighted peer loss [30]. Peer: peer loss [21].

| Dataset | $e_a$ | $e_b$ | Metrics | LESS NOISE | | MORE NOISE | |
| | | | | LR | GPL | LR | Peer |
|---|---|---|---|---|---|---|---|
| Adult | 0.1 | 0.3 | *accuracy* | 72.57 | 71.92 | 71.07 | 73.21 |
| | | | *fairness* | 2.37 | 3.39 | 1.83 | 1.95 |
| | 0.2 | 0.3 | *accuracy* | 72.4 | 72.92 | 73.07 | 71.8 |
| | | | *fairness* | 6.67 | 3.36 | 4.21 | 0.93 |
| | 0.2 | 0.4 | *accuracy* | 72.73 | 71.2 | 71.88 | 73.02 |
| | | | *fairness* | 6.48 | 2.95 | 3.16 | 1.67 |
| | 0.3 | 0.4 | *accuracy* | 73.15 | 73.74 | 71.36 | 72.74 |
| | | | *fairness* | 5.29 | 4.11 | 5.49 | 1.88 |
| Compas | 0.1 | 0.3 | *accuracy* | 63.88 | 63.73 | 64.56 | 64.33 |
| | | | *fairness* | 7.17 | 6.58 | 7.35 | 1.89 |
| | 0.2 | 0.3 | *accuracy* | 63.73 | 63.28 | 64.26 | 67.8 |
| | | | *fairness* | 10.52 | 4.47 | 7.10 | 2.76 |
| | 0.2 | 0.4 | *accuracy* | 62.60 | 66.03 | 66.22 | 64.15 |
| | | | *fairness* | 2.87 | 7.55 | 6.07 | 3.63 |
| | 0.3 | 0.4 | *accuracy* | 61.93 | 62.08 | 61.63 | 62.68 |
| | | | *fairness* | 17.97 | 3.06 | 7.70 | 3.74 |
| Fairface | 0.2 | 0.4 | *accuracy* | 86.97 | 87.47 | 88.19 | 87.93 |
| | | | *fairness* | 5.87 | 4.70 | 1.38 | 0.25 |
| | 0.1 | 0.3 | *accuracy* | 88.23 | 88.23 | 88.58 | 88.60 |
| | | | *fairness* | 5.53 | 4.93 | 2.11 | 2.17 |
| | 0.0 | 0.2 | *accuracy* | 88.61 | 88.53 | 88.90 | 88.85 |
| | | | *fairness* | 4.05 | 3.75 | 2.64 | 2.20 |
| | 0.0 | 0.1 | *accuracy* | 89.08 | 88.84 | 89.00 | 89.05 |
| | | | *fairness* | 3.99 | 3.92 | 2.97 | 2.91 |
| | 0.2 | 0.3 | *accuracy* | 88.63 | 88.78 | 88.80 | 88.83 |
| | | | *fairness* | 3.50 | 3.14 | 2.19 | 1.33 |