# OpenReview forum: "Can Less be More? When Increasing-to-Balancing Label Noise Rates Considered Beneficial"
_NeurIPS.cc/2021/Conference — NeurIPS 2021 Poster_

### Official Review · Reviewer_JhxR · 2021-07-14

**Rating:** 7
**Confidence:** 2

**Summary:**

This submission studies the paradigm of injecting label-noise from the perspective of fairness and robustness. The motivation stems from the fact that balancing the label proportions yields an easier learning problem and improves fairness guarantees. The submission derives a bound between the corrected loss with label noise and the population risk, which through the use of standard concentration inequalities, involves the Rademacher complexity. In order to fix one, they propose a method that detects the right choice of labels that suffer from noise, which then is turned into an algorithm for robust and fair label correction.

**Limitations And Societal Impact:**

Yes they have

**Main Review:**

The main contributions of the paper consist of two major components: (1) elucidating the trade-off and an in-depth analysis of the error rates in classification along with a generalization bound to explicitly outline this and (2) a detection method that allows you to determine the right group to inject noise for further processing.

The main strength of the submission is indeed (1), which presents an interesting analysis that explains this trade-off and serves an extremely useful purpose to both the practitioner and theoretician working in fairness and robustness. The contribution (2) is more preliminary at this point however will serve useful for future developments that incorporate the results of this paper.

The weaknesses I can gather include the tightness of the bound presented in (1). Since it involves the Rademacher complexity, it leads me to believe this may be inaccessible to conventional learning settings where we use deep neural networks. I am not entirely familiar with this literature and am not sure whether this as a starting point is substantial enough.

The submission is very clear with its contribution and makes it easy for the reader to understand and follow the story. The notation is mostly easy to follow and uses standard conventions throughout - small things like Rademacher complexity is not formally defined however this is no big deal as it is not the focus of this result.

Based on my knowledge, it seems the related work is discussed appropriately and the quality of correctness is sufficient throughout the paper as I was not able to find any mistake in the derivation of the results. I believe the topics of dealing at the intersection of robustness, fairness and label-noise is extremely relevant to the NeurIPS community.

Overall, I think this paper meets the bar of inclusion for NeurIPS due to the importance of the problem, elucidation of this trade-off between noise rates and relatively simple method of group-detection for downstream tasks. The results in this work would greatly benefit the machine learning community.

Some questions/comments for the authors:

(1) Why is sgn used in Line 123 in Equation (2)? It seems to me that in the binary classification case it is redundant since Y is either +1 or -1 or perhaps I'm not appreciating some generality here?

(2) It would be useful if you could explain some intuition behind the corrected loss since a non-expert finds it difficult to grasp the weighting parameters.

(2) A minor weakness I pointed out is the Rademacher complexity involvement in the bound - I was wondering if you could have other complexity measures appearing here perhaps if you change the analysis. For example, in the discrete setting, would it be possible to have the VC dimension appearing or even a PAC-Bayesian analysis? I tried to replicate the arguments however it was not so clear to me if the Rademacher complexity can be avoided.

----------------------------------------------------------------------------------------------------------------------------------------
I have read the other reviews and rebuttals. The authors have addressed my questions and concerned and so I will be increasing my score.

**Time Spent Reviewing:**

10

---

> ### Author Response · Authors · 2021-08-10
> **Rebuttal to feedback from Reviewer JhxR**
>
> We thank the reviewer for raising the questions (we particularly think the question regarding the complexity bound merits a future study). We hope our rebuttal can help alleviate some of them:
>
> **Q: Rademacher complexity**
>
> We completely agree that comparing bounds can be vacuous. It is certainly a motivation for our empirical verifications and noise balance effort. A tighter bound will always be helpful. We will clarify.
>
> The reviewer also raised a good point of using Rademacher complexity specifically. Most of our analysis can extend as long as we are offered a regret bound as a function of the label noise rate. For instance, Rademacher complexity can be bounded using VC-dimension under a variety of problem setups. For instance, see Example 2 of the lecture note: https://www.cs.cmu.edu/~ninamf/ML11/lect1117.pdf. (the Radermacher complexity can be further bounded by its empirical version - they mostly differ by another convergence term)
>
> For another example on the generalization error of a deep neural network: indeed the generalization of a deep neural network requires different analysis. We noted a recent paper that looked into the generalization error when a neural network can memorize noisy training labels [1]. It’s possible to borrow results from the paper to tailor our claim.
>
> **Q: "(1): Why is sgn used in Line 123 in Equation (2)? It seems to me that in the binary classification case it is redundant since Y is either +1 or -1 or perhaps I'm not appreciating some generality here?"**
>
> Since we use $e_-$ and $e_+$ to denote the noise rates, we use the sign function to transform labels in {-1, +1} into {-, +}.
>
> **Q: "(2): It would be useful if you could explain some intuition behind the corrected loss since a non-expert finds it difficult to grasp the weighting parameters."**
>
> Thanks for your suggestions!  The corrected loss uses the parameter to balance the two possibilities that the noisy label can differ from the true one. Then, when taking expectation wrt the randomness of the noisy labels $\tilde Y|Y$, the corrected loss with true noise parameters defined in Equation (2) is an unbiased estimate of the original loss on the clean labels. We would add more descriptions in the revised version of the paper.
>
>
> [1] Understanding Instance-Level Label Noise: Disparate Impacts and Treatments, Yang Liu, International Conference on Machine Learning (ICML), 2021.

---

### Official Review · Reviewer_K19n · 2021-07-15

**Rating:** 6
**Confidence:** 4

**Summary:**

In this work, the authors aim to show that adding label noise in some cases could be beneficial to the test performance and fairness of models. Firstly, they find that increasing-to-balancing can result in an easier learning problem and improves fairness guarantees against label bias. Then they propose a method by inserting label noise to the group of labels with lower noise rates. To estimate the noise rates of various groups, the authors propose a detection method by checking the agreements of noisy labels among local neighbors. Experiments are conducted on three tiny datasets and CIFAR-10 dataset across different levels of noise rates to verify the effectiveness of the proposed algorithm.


**Limitations And Societal Impact:**

The authors have addressed the limitations in noise types. It would be better if the authors can add more large datasets to show the applicability of the proposed method.



**Main Review:**

Pros:

As far as I’m aware, this work is the first to consider model fairness in the setting of label noise. In this work, the authors show that increasing-to-balancing can improve the performance under the setting of asymmetric noisy labels, which is somewhat counter-intuitive from the view of noisy rate.

This paper is well-written and easy-to-follow. The authors also describe the limitation of this work in the aspect of noisy types.


Cons:
In the setting of asymmetric noisy labels, it seems that the gap of noisy rate among classes can be treated as a class-imbalance problem. In the view of class imbalance, the observation would be not surprising because increasing-to-balancing could alleviate the problem of class imbalance. Then the trade-off becomes reducing the noise rate or the class imbalance rate, which reduces the contribution of this work.

The proposed detection method is similar to the KNN method in [1] because the agreement strategy among local neighbors can be seen as a variant of KNN.

The three datasets used in experiments are very small. Although CIFAR-10 is also used, they only train with the image features extracted by a pre-trained vision transformer. Therefore, the experiments are not enough to show the conclusion from observations and the proposed method can extend to some large datasets trained from scratch, like MNIST, CIFAR-10/100 and Clothing1M (imbalanced).

The improvements from the proposed method are weak, especially on the CIFAR10 dataset. Although improving test accuracy is not the only purpose of this work, the weak improvements are still not convincing enough.


Other comments:

Why don’t try reducing label noise of the groups of labels with higher noise rate by sample selection?  Then we don’t need to consider the tradeoff because we reduce both the noise rate and imbalance rate.

[1] Bahri, Dara, Heinrich Jiang, and Maya Gupta. "Deep k-nn for noisy labels." International Conference on Machine Learning. PMLR, 2020.


***

My concerns have been addressed by the response, thus I decide to improve my score to 6. Although the improvements are still somewhat weak in some cases, this work provides new insights about asymmetric noisy labels for the community, which are more important.

**Time Spent Reviewing:**

6

---

> ### Author Response · Authors · 2021-08-10
> **Rebuttal to feedback from Reviewer K19n**
>
> We thank the reviewer for raising questions about our contribution. Clarifying them indeed helps us better position our technical results in the literature:
>
> **Q: "In the setting of asymmetric noisy labels, it seems that the gap of noisy rate among classes can be treated as a class-imbalance problem."**
>
> First of all, though relevant, we argue that the gap of noisy rate among classes is not equal to a class-imbalance problem. The balance of class on the noisy data depends also on the prior of the class distribution on the clean data ($\mathbb P(Y=+1), \mathbb P(Y=-1)$). For instance, when true label distribution is $\mathbb P(Y=+1) = 0.3, \mathbb P(Y=-1) =0.7$, and initial error rates are $e_+ = 0.1, e_- = 0.2$ we have
> \begin{align}
> \mathbb P(\tilde{Y}=+1) &= \mathbb P(Y=+1)\cdot \mathbb P(\tilde{Y}=+1|Y=+1) + \mathbb P(Y=-1)\cdot \mathbb P(\tilde{Y}=+1|Y=-1) \\\\
> &=\mathbb P(Y=+1)\cdot (1-e_+) + \mathbb P(Y=-1)\cdot e_-\\\\
> &= 0.3 \cdot 0.9 + 0.7 \cdot 0.2 = 0.41.
> \end{align}
> And  $\mathbb P(\tilde{Y}=+1) = 0.59.$
>
> When balancing noise rate $e_+ = e_- = 0.2$ we have the following noisy label distribution
> \begin{align}
> \mathbb P(\tilde{Y}=+1) &= \mathbb P(Y=+1)\cdot \mathbb P(\tilde{Y}=+1|Y=+1) + \mathbb P(Y=-1)\cdot \mathbb P(\tilde{Y}=+1|Y=-1) \\\\
> &=\mathbb P(Y=+1)\cdot (1-e_+) + \mathbb P(Y=-1)\cdot e_-\\\\
> &= 0.3 \cdot 0.8 + 0.7 \cdot 0.2 = 0.38.
> \end{align}
> And  $\mathbb P(\tilde{Y}=+1) = 0.62$ - indeed the noisy label distribution becomes less balanced even though we have balanced the noise rates. Therefore balanced noise rates do not imply class balance, and neither does class balance imply balanced noise rates.
>
> **Q: "The proposed detection method is similar to the KNN method in [1] because the agreement strategy among local neighbors can be seen as a variant of KNN."**
>
> Thanks for pointing out the relevant work. We will reference.
>
> Using the K-NN approach would be an interesting proposal but we often find the K-NN requirements can be challenged, especially for a smaller set of data with limited features. In contrast, our approach requires only 2-NN clusterability (Section 5, Definition 7, line #207), that is each data shares the same true label with its two nearest neighbors. This is a much easier requirement to satisfy.
>
> The idea of checking consensus patterns (Section 5, Definition 8) also differs from the K-NN paper that primarily checks the majority votes among neighboring instances: our idea is to measure how each noisy label agrees with its two nearest neighbors ($\mathbb P(\tilde{Y}_2 = \tilde{Y}_3 = +1|\tilde{Y}_1 = +1$ and $\mathbb P(\tilde{Y}_2 = \tilde{Y}_3 = -1|\tilde{Y}_1 = -1$) and use the measure to determine the noisier class. Checking the agreement patterns with both closest neighbor instances is new, and is entirely different from the K-NN approach. This detection procedure described in Figure 3 in Appendix A.11 has strong theoretical guarantees (see Section 5, Theorem 9, line#219) - this is also a new finding to the community. In all, we believe our proposal differs substantially from the K-NN paper.
>
> **Q: "The three datasets used in experiments are very small."**
>
> Serving heavy and large datasets specifically is not the main purpose of this paper. We also wanted to test on datasets that have fairness implications. The selected UCI and Fairface datasets were believed to be better fits. Nonetheless, we think it is unfair to call these datasets as being “very small”.  The Adult dataset has 48,842 examples. The Fairface has 108,501 examples. In comparison, CIFAR-10 and CIFAR-100 have 50,000 training examples and 10,000 test examples. In addition, we conducted experiments for training from scratch on MNIST datasets. The results are as follows:
>
> | gap | Mis.SL | Est.SL | CE | Peer | CE (Noise+) | Peer (Noise+) |
> | --- | ---- | ------- | --- | --- | --- | --- |
> |  0.1 | $89.59\pm0.01$ | $89.69\pm0.07$ | $86.66\pm0.54$ | $88.12\pm0.01$ | $86.81\pm0.62$ | $89.19\pm0.05$|
> | 0.2 | $88.10\pm0.10$ | $88.61\pm0.16$ | $84.53\pm1.60$ | $87.21\pm0.53$ | $85.97\pm0.69$ | $89.12\pm0.24$ |
> | 0.3 | $84.97\pm0.11$ | $86.88\pm0.17$ | $85.24\pm1.05$ | $86.35\pm0.33$ | $81.89\pm1.54$ | $88.75\pm0.19$ |
>
> As shown, the increasing-to-balance methods consistently achieves a higher accuracy when noise gap is lower than 0.2. When the noise gap is as high as 0.3, we find that the cross entropy with increasing-to-balancing doesn’t outperform the vanilla cross entropy, but peer loss with increasing-to-balancing still has the highest accuracy. This observation aligns with our observations on the CIFAR-10 dataset.
>
> Due to limited time and computing resources, we performed preliminary study on Clothing1M. For a fair comparison, we adopted the same setting as described in [2] and trained a ResNet-50 classifier. The table below compares our methods with some other baselines. Without a careful tuning of training parameters, vanilla CE with Noise+ reached 70.37% test accuracy. In comparison, standard CE achieves 68.94% (we get the number from [1]), loss correction achieves 69.84% [2], Co-Teaching is 70.15% [3]. We understand there have been higher numbers reported, but we hope that with fine tunes and by applying Noise+ with peer loss and other approaches, we will be able to reach an even higher performance.
>
> | Method | Test Accuracy |
> | --- | --- |
> | CE | 68.94% |
> | Loss Correction [2] | 69.84% |
> | Co-Teaching [3] | 70.15% |
> | CE + Noise+ | 70.37% |
>
> **Q: "Improvements are small."**
>
>  We want to note that the loss correction and peer loss baselines are already strong baselines documented in the literature. But we still consistently observe a better performance. Our improvements compared to weaker baselines (using cross-entropy loss) are quite substantial (close to 10%) when the noise rate is high ($e_-=0.2, e_+=0.4$), while we do expect big performance differences when noise rates are low. Our approach (particularly when paired with peer loss) also achieves consistently low fairness violations across different dataset (Table 3 in the main paper and Table 6 in the appendix), which is an equally important focus of this study.
>
> **Q: "Why don’t try reducing label noise of the groups of labels with higher noise rate by sample selection? Then we don’t need to consider the tradeoff because we reduce both the noise rate and imbalance rate."**
>
> To preface our response, we want to highlight that we aim for a data processing process that 1) is easy to implement, 2) requires as few assumptions as possible, 3) doesn’t require ground truth labels and 4) returns us strong theoretical guarantees for accuracy and fairness.
>
> Decreasing the noise rate is an interesting idea. It was certainly the first thing that came to our mind to balance the noise rate. If this can be easily done, we agree it would be ideal, and indeed we will enjoy the best of the two worlds. First, we want to clarify that both increasing and decreasing procedures would require the detection procedure, so our contribution for the detection step is independent.
>
> Now on to whether we shall increase or decrease the noise level. Reduce-to-balancing the noise rate is a much harder procedure to develop and implement. First of all, it will require a credible noisy label detection procedure. We checked the literature, and this detection process would require either 1) structural assumption about training a decent model to help detect the corrupted labels or 2) the K-NN clusterability assumption, which leads to unclear analytical results in our setting (particularly when we consider the fairness constraints).
>
> The above technical challenge is due to the fact that after detecting the noise labels, we will need a removal procedure to carefully balance the decrease of error rates of both label classes. We found this to be a much more complicated procedure, which requires understanding the likelihood of detection to be wrong (again by itself a challenging question, without having access to ground truth labels), so to evaluate how much a flip would be preferred, and to guarantee the detection is not introducing more asymmetricity between the two label classes. If we were able to detect the wrong labels confidently, we could probably reduce both classes’ errors to 0 and overcome the label noise problem. Nevertheless, we feel this is a bit too optimistic and don’t find such a solution framework without substantial and styled assumptions.
>
> While being an interesting future direction to explore, we believe reducing noise rate would be hard to establish theoretical guarantees for both accuracy and fairness. In particular, though it is possible (empirically) that the accuracy can be improved without a meticulous procedure of removing noisy labels, it is hard to guarantee so regarding the fairness constraints.
>
> In contrast, following this detection step, our Noise+ procedure is light and easy to implement and can be done fast, without relying on any outputs from the learning problem itself. Noise+ serves as a preprocessing tool that can easily be implemented within existing solutions and has provable guarantees, which are the more demanding criteria for reducing noise rates. We believe the Noise+ solution has a broad application when there is a need to preprocess the data to balance the noise rates.
>
> [1] https://paperswithcode.com/sota/image-classification-on-clothing1m
>
> [2] Patrini, G., Rozza, A., Menon, A., Nock, R., & Qu, L. (2017). Making Deep Neural Networks Robust to Label Noise: A Loss Correction Approach. 2017 IEEE Conference on Computer Vision and Pattern Recognition (CVPR), 2233-2241.
>
> [3] Han, B., Yao, Q., Yu, X., Niu, G., Xu, M., Hu, W., Tsang, I., & Sugiyama, M. (2018). Co-teaching: Robust training of deep neural networks with extremely noisy labels. NeurIPS.

---

> > ### Comment · Reviewer_K19n · 2021-08-13
> > **Thank you for the additional details!**
> >
> > The additional discussions have addressed my concerns. Although the proposed detection method is similar in spirit to the K-NN method, I agree that the proposed method is much easier to be implemented and does not require an additional clean validation dataset.

---

### Official Review · Reviewer_A8eu · 2021-07-18

**Rating:** 7
**Confidence:** 3

**Summary:**

This paper explores ways to improve the generalization performance and fairness guarantees, by increasing a certain class of instances' label noise to balance the rates between different classes. Theorem suggests that when the gap between noise rates is small and when we don't have high confidence in estimating the noise rates, it may be beneficial to increase noise rates. The paper further proposes a method to insert label noise without knowing the ground truth training labels. Experiments show that the rate balancing procedure can be beneficial to increase test performance.

**Ethical Concerns:**

No comments.

**Limitations And Societal Impact:**

The paper discusses that the method is limited for class-conditional noise. It does not directly discuss societal impacts, but fairness guarantees is discussed throughout the paper.

**Main Review:**

Pros:
- The theorems shown in the paper are interesting, and provides new insights.
- The paper is well organized, starting from theoretical discussions and then verifying them through experiments.

Cons:
- Although the theoretical contributions are interesting, some practical limitations remain: the noise gap needs to be small in order for the noise equalization to be beneficial in the equation in line 152~153 and also shown in experiments in Table 2. Is it possible to perform similar large gap experiments with the binary classification setup? Currently, the largest gap seems to be 0.2 in Table 1. It would make the paper stronger if we can see consistent results across different datasets. I was also curious if it is still beneficial under a setting with high noise rates but low noise gap.

Other minor comments:
- It would also be interesting to see experiments without using a pre-trained model for Table 2. Training from scratch with noisy labels may have more significant effects on the trained model.
- The gamma parameter was fixed in the experiments, but how sensitive are the results w.r.t. gamma?
- line 126: denonminator --> denominator
- line 178: FRP --> FPR?
- Reference [27] and [28] are the same

=======================================
After author response: Thank you for the additional experiments and for answering my questions. Since I do not have any further major concerns, I would like to raise my score to 7. Additional minor comment: typo on agreeing" on line 215.

**Time Spent Reviewing:**

4

---

> ### Author Response · Authors · 2021-08-10
> **Rebuttal to feedback from Reviewer A8eu**
>
> We thank the reviewer for raising detailed questions about our experiments - we will definitely add them to the next version of our work.
>
> We respond below:
>
> **Q: "Is it possible to perform similar large gap experiments with the binary classification setup?"**
>
> For binary classification, the noise rate for either class has to be bounded by 0.5 - otherwise, the training data is considered contaminated and it’s hopeless to learn things useful. Within the 0.5 bounds, we think a 0.2/0.3 gap is reasonable. We have conducted experiments with gap 0.3 in Table 4 in the Appendix B, and our results are consistently good. We are sorry we didn’t put in the main paper due to the space limitations.
>
>
> **Q: "I was also curious if it is still beneficial under a setting with high noise rates but low noise gap."**
>
> Our method is still beneficial when noise rates are high but noise gap is low. We complement an additional run with noise rates 0.3 and 0.4 (0.1 gap) on the Adult dataset, and report results here:
>
> | $e_-$ | $e_+$ | Mis.SL | Est.SL | CE | Peer | CE (Noise+) | Peer (Noise+) |
> | --- | ---- | ------- | --- | --- | --- | --- | --- |
> | 0.3 | 0.4 | $52.65\pm0.53$ | $72.67\pm0.26$ | $71.55\pm0.88$ | $73.49\pm0.18$ | $72.54\pm1.84$  | $74.27\pm0.20$ |
>
>
> **Q: "It would also be interesting to see experiments without using a pre-trained model for Table 2"**
>
> We conducted experiments for training from scratch on MNIST datasets. The results are as follows:
>
> | gap | Mis.SL | Est.SL | CE | Peer | CE (Noise+) | Peer (Noise+) |
> | --- | ---- | ------- | --- | --- | --- | --- |
> |  0.1 | $89.59\pm0.01$ | $89.69\pm0.07$ | $86.66\pm0.54$ | $88.12\pm0.01$ | $86.81\pm0.62$ | $89.19\pm0.05$|
> | 0.2 | $88.10\pm0.10$ | $88.61\pm0.16$ | $84.53\pm1.60$ | $87.21\pm0.53$ | $85.97\pm0.69$ | $89.12\pm0.24$ |
> | 0.3 | $84.97\pm0.11$ | $86.88\pm0.17$ | $85.24\pm1.05$ | $86.35\pm0.33$ | $81.89\pm1.54$ | $88.75\pm0.19$ |
>
> As is shown, the increasing-to-balance methods consistently achieves a higher accuracy when noise gap is lower than 0.2. When the noise gap is as high as 0.3, we find that the cross entropy with increasing-to-balancing doesn’t outperform the vanilla cross entropy, but peer loss with increasing-to-balancing still has the highest accuracy. This observation aligns with our observations on the CIFAR-10 dataset.
>
> The pre-trained model was mostly for extracting features for image datasets, which becomes more or less a standard. We do observe that using a pre-trained model can extract features that satisfy our clusterability assumptions. To our best knowledge (but please correct us if we are wrong), most CIFAR experiments use certain pre-trained neural networks to perform feature extractions. The UCI data wouldn’t need any of the pre-trained models.
>
>
> **Q: "The gamma parameter was fixed in the experiments, but how sensitive are the results w.r.t. gamma?"**
>
> We tried $\gamma$ values at different scales and the $\gamma$ parameters we set in the experiments have a better trade-off. If $\gamma$ is smaller, it might need several runs to get a feasible solution. When $\gamma$ is between 0.1% - 1.0%, we usually obtain a near-optimal solution and results are not sensitive in this range. In practice, we indeed observe that the noise rates after our Noise+ procedure with $\gamma$ between 0.1% and 1.0% are near balanced and the classifiers trained later on generall have a quite evident improvement on fairness and accuracy metrics. When $\gamma$ is larger than 5%, it is very hard to get balanced noise rates after the Noise+ program.
>
>
> **Other minor comments**
>
> Thank you for pointing out the typos! We will revise them in the newest version.

---

### Official Review · Reviewer_ayxB · 2021-07-19

**Rating:** 6
**Confidence:** 4

**Summary:**

This paper considers classification in the case of noisy training data, where one subset of points suffers from higher label noise then the complementary subset. This paper claims that by increasing the label noise rate on the less noisy subset, it is possible to obtain more accurate models since the learning problem is easier.

Additionally, in the context of fairness, increasing the noise rate of the advantaged group improves fairness guarantees in the sense that when we are fair on the noisy data, we are also fair on the clean data.

Lastly, this paper contributes a method to detect which subset suffers from higher noise rates (and how much more is suffers) without access to the ground truth.

**Limitations And Societal Impact:**

Regarding limitations, I have addressed this in points 2, 3, and 4, but I would be glad about clarifications from the authors.

**Main Review:**

Main Points

1) Observation 2 states that the learning problem becomes easier. In what way? In fact, later on, we establish that we actually have an increase in learning difficulty while overall we reduce the upper bound on the excess risk.

2) An important point in Theorem 1 seems to be that the noise rates are not known and we use estimated noise rates to correct the loss. This adds some misspecification term in our bound, which is at the core of the tradeoff. This could be stated more upfront in the beginning since the main problem seems to be how to obtain accurate estimates of the noise rates. However, when are the estimated error rates important later on in the experiments? Is this loss correction (2) always applied?  The peer loss seems to specifically remove the limitation of needing the rates. Therefore, the benefit elaborated in Section 3 seems irrelevant here (but is still true for fairness).

3) Furthermore, I dont see why a lower upper bound actually means that a learning problem is 'easier'. Since we are looking only at upper bounds it is well possible that the excess risk of the balanced noise case is still higher than the excess risk in Theorem 1. I would see this observation more as a motivation to experiment with balanced noise which is indeed emperically advantageous as seen in the experiments.

4) Overall, I would suggest to communicate the meaning of Section 3 differently. It seems to depend heavily on the loss corretion approach and estimated noise rates, which later on become irrelevant. I think this theoretical observation is nice to motivate the contributed algorithm. After all, Noise+ still improves the baseline Peer loss, even though the peer loss does not need estimates of the noise rates.

5) Proof of Lemma 1: Why does it hold that P(h(X) \neq Y^hat | Y^hat \neq Y) =  P(h(X) = Y| Y^hat \neq Y) = P(h(X) = Y)? The last step is not trivial, even though we know that the label noise is independent of X.

6) Regarding condition 2 in Theorem 3. How strong is it and what does it imply? We are looking at the optimal classifier on the clean D and the optimal classifier learned with newly generated noise labels. Those two are now flipped and evaluated on the original noisy data. Just intuitively, I would think that h tilde would have a lower loss on the noisy data than h diamond. And if you flip those, you are in the case of condition 2). Is that accurate?

7) It is unclear to me what the competing methods are. What is the difference between MisSL and EstSL? Earlier in the paper, when estimated versions of the noise rates were used, they were called 'model mis-specification error'.

Minor Points:

9) Line 143: The expected value is written with tildes. Is that correct? I think, here we are talking about the noise balanced case notated with hats.

10) Notation is sometimes confusing. h with tilde is defined with y hats and  and h hat is defined with y tildes. If you would rename the loss l^hat, you wouldnt need a hat in h, but could put the loss name in the subscript.

11) Lines 245 to 259 are very condensed and confusing. Is the notation about C_l and C_r established at that point?

12) What does Lemma 5 tell me? Simply that I need to at least estimate the noise rates right? And with Theorem 6, is this lemma even relevant?

13) For more clarity, it would be good to shortly define what a calibrated loss function is.

14) The baseline methods, the noise procedure, and the peer loss need more description. This would be a helpful addition to the supplement.

**Time Spent Reviewing:**

10 hours

---

> ### Author Response · Authors · 2021-08-10
> **Rebuttal to feedback from Reviewer ayxB**
>
> We thank the reviewer for the detailed comments. Clarifying them has greatly improved our presentation. The suggested changes are very well received.
>
> **1. "Why does the learning problem become easier?"**
>
>
> Sorry for the confusion - by “the learning problem becomes easier when the noise rates are balanced” what we really meant is the associated computational ease. We didn’t imply the learning difficulty defined using the achieved regret: in the best scenario, having a higher noise rate would unavoidably be a higher regret. When error rates are balanced, as we show in the paper, invoking loss correction procedures that require the specifications of noise rates becomes unnecessary. (Note that the bound the reviewer refers to is derived by applying the standard risk minimization without invoking any loss correction procedure)
>
> Broadly in the literature of learning with noisy labels, the loss functions satisfying the symmetry conditions are noise tolerant when the noise rates are balanced (see Theorem 1 in [1]). That means directly minimizing a symmetric loss, such as 0-1 loss and hinge loss, on the noisy data without any noise correction could be equivalent to minimize the loss on the clean data distribution. In this way, the learning problem is much easier.
>
>
>
> **2. "However, when are the estimated error rates important later on in the experiments? Is this loss correction (2) always applied?"**
>
> The loss correction is not applied to peer loss functions baseline but has been applied in others: MisSL and EstSL, which are loss correction using mis-specified and estimated noise rates respectively ([2]). The reviewer was correct that the benefits of balancing the errors were mostly motivated by approaches that explicitly use the error rates. We would like to further note a couple of facts: 1) peer loss’s strong performance guarantee was established for balanced prior. When it’s not, the performance of peer loss degrades. Empirically we observe that our noise balancing procedure benefits peer loss too. (as observed by the reviewer too). We start to think this balancing procedure may balance other approaches that do not use noise rates (of course many of them do not have consistency guarantees so we haven’t delved into many of them) 2) As the reviewer correctly pointed out, even for approaches that don’t use error rates, handling the fairness constraints is non-trivial.
>
> We will be more upfront of this observation in the next version.
>
> **3. Question on comparing bounds**
>
> Agree. Comparing bounds can be vacuous. It is certainly a motivation for our empirical verifications. We will clarify.
>
> **4. Suggestion for communicating Section 3**
>
> This is well received. Thanks.
>
> **5. "Why does it hold that P(h(X) \neq Y^hat | Y^hat \neq Y) = P(h(X) = Y| Y^hat \neq Y) = P(h(X) = Y)?"**
>
> Assuming the reviewer meant Lemma 2: Note that Lemma 2 is established for the symmetric error rates: so tilde Y is independent of both label classes (cause it’s the same for both classes Y=-1 or +1) and X. Some detailed derivations:
>
> $\mathbb P(h(X) \neq \hat{Y} | \hat{Y} \neq Y) =\mathbb P(h(X) \neq \hat{Y} , \hat{Y} \neq Y| \hat{Y} \neq Y)$  In binary cases, $h(X) \neq \hat{Y} , \hat{Y} \neq Y$ implies $h(x) = Y$. so $\mathbb P(h(X) \neq \hat{Y} | \hat{Y} \neq Y) = \mathbb P(h(X) = Y| \hat{Y} \neq Y)$.
>
> $\mathbb P(h(X) = Y| \hat{Y} \neq Y)$ can be expressed as $\mathbb P(h(X) = Y, \hat{Y} \neq Y) / \mathbb P(\hat{Y} \neq Y) =\mathbb  P (h(X) = Y) \mathbb P(\hat{Y} \neq Y) /\mathbb  P(\hat{Y} \neq Y) = \mathbb P(h(X) = Y)$.
>
> **6. "Regarding condition 2 in Theorem 3. How strong is it and what does it imply?"**
>
> The reviewer is correct. The condition is saying that flipping the prediction of the optimal classifier trained using the noisy data, leads to worse or no better empirical risk than flipping $h^{\diamond}$. This condition is often satisfied for binary classification - if one classifier performs the best on the empirical data, flipping its prediction often results in a very wrong one.
>
> **7. "It is unclear to me what the competing methods are. What is the difference between MisSL and EstSL?"**
>
> The Mis.SL is using the loss correction method Eq (2) with mis-specified noise parameters (given as parameters of the experiments), while Est.SL is using the loss correction with estimated noise parameters (using the confident learning procedure [2]). Often Mis.SL incurs a higher mis-specification error than Est.SL. And as we see in the results, Est.SL has a higher accuracy than Mis.SL in general.
>
> **9. "Line 143: The expected value is written with tildes."**
>
> Sorry, this was a typo. It should be $\hat{Y}$. Thanks for pointing it out!
>
> **11. "Lines 245 to 259 are very condensed and confusing. Is the notation about C_l and C_r established at that point?"**
>
> The notation $C_l$ is corresponding to the agreement gap (PA - NA) with the flipping parameter $\epsilon_l$, and the notation $C_r$ is corresponding to the agreement gap (PA - NA) with the flipping parameter $\epsilon_r$. We will add the exact definitions.
>
>
> **12. "What does Lemma 5 tell me? Simply that I need to at least estimate the noise rates right? And with Theorem 6, is this lemma even relevant?"**
>
> Lemma 5 establishes the lower bounds between the true TPR and the corrected TPR with estimated noise parameters, and shows the fairness violation when the estimations of noise rates are inaccurate. Theorem 6 states that after we balance the error rates, we won’t need to concern about the errors in estimating $e_+,e_-$ since equalizing the constraints on the noisy data suffices to equalize the true fairness constraints, per Lemma 4 (we dropped a tilde on top of the last TPR in Eqn 5. sorry)
>
> Thank you for the other notational and clarification suggestions.
>
> [1] Ghosh, A., Manwani, N., & Sastry, P. (2015). Making risk minimization tolerant to label noise. Neurocomputing, 160, 93-107.
>
> [2] Northcutt, C.G., Jiang, L., & Chuang, I.L. (2021). Confident Learning: Estimating Uncertainty in Dataset Labels. J. Artif. Intell. Res., 70, 1373-1411.

---

> > ### Author Response · Authors · 2021-08-30
> > **Any feedback or further discussions on this work?**
> >
> > Dear Reviewer ayxB,
> >
> > Since we haven't received any updates or feedback, we want to kindly follow up with you that whether we have addressed all your concerns or you still have any further questions that we could clarify? We are looking forward to further discussions. Thank you very much!
> >
> > Sincerely,
> >
> > Authors of paper 3820

---

### Decision · Program_Chairs · 2021-09-27

**Decision:**

Accept (Poster)

**Comment:**

Reviewers were unanimously positive about the paper's contribution to a topical problem, namely, learning with label noise and its implications on model fairness. The initial reviews raised some technical questions that were satisfactorily addressed by the authors. The authors are further encouraged to incorporate the other suggestions provided by reviewers for the updated version of the work.